# From subcritical behavior to a correlation-induced transition in rumor models

Guilherme Ferraz de Arruda [1✉], Lucas G. S. Jeub [1], Angélica S. Mata [2], Francisco A. Rodrigues[3] & Yamir Moreno [1,4,5]

Rumors and information spreading emerge naturally from human-to-human interactions and have a growing impact on our everyday life due to increasing and faster access to information, whether trustworthy or not. A popular mathematical model for spreading rumors, data, or news is the Maki–Thompson model. Mean-field approximations suggested that this model does not have a phase transition, with rumors always reaching a fraction of the population. Conversely, here, we show that a continuous phase transition is present in this model. Moreover, we explore a modified version of the Maki–Thompson model that includes a forgetting mechanism, changing the Markov chain's nature and allowing us to use a plethora of analytic and numeric methods. Particularly, we characterize the subcritical behavior, where the lifespan of a rumor increases as the spreading rate drops, following a power-law relationship. Our findings show that the dynamic behavior of rumor models is much richer than shown in previous investigations.

[1] ISI Foundation, Via Chisola 5, 10126 Torino, Italy. [2] Departamento de Física, Universidade Federal de Lavras, 37200-900 Lavras, Minas Gerais, Brazil. [3] Departamento de Matemática Aplicada e Estatística, Instituto de Ciências Matemáticas e de Computação, Universidade de São Paulo - Campus de São Carlos, Caixa Postal 668, 13560-970 São Carlos, SP, Brazil. [4] Institute for Biocomputation and Physics of Complex Systems (BIFI), University of Zaragoza, Zaragoza 50009, Spain. [5] Department of Theoretical Physics, University of Zaragoza, Zaragoza 50009, Spain. ✉email: gui.f.arruda@gmail.com

Rumor spreading is a spontaneous process that emerges from social interactions and can occur in real or virtual environments. Electronic media has increasingly contributed to the impact of this type of process on people's daily lives. More concretely, these processes include, but are not limited to, propagation of information or gossips and can even be used to model fake news[1–5]. However, the relevance of rumor models is not limited to social context. Indeed, this class of models inspired and constituted the theoretical basis for the so-called gossip protocol[6], which is a powerful paradigm used in the design of decentralized distributed protocols that are reliable and efficient[7]. The applications of this protocol are quite wide, including Peer-to-Peer Networks (P2P)[7–9] and specific tasks as failure detection[7,10], to implement garbage collection[7,11], to compute aggregate information[7,12], and to allocate resources[7,13]. Moreover, specific real-life examples, include the Gnutella P2P Network[9] and cryptocurrencies networks as Bitcoin[14,15] or Ethereum[16] and their derivations. Although these examples are easily relatable to our daily lives, surprisingly, this class of processes is much less explored than other stochastic dynamics[17,18].

Due to its practical relevance, it is essential to understand the evolution mechanism of rumor spreading on heterogeneously connected populations. The first rumor model was introduced in 1964 by Daley and Kendall (DK)[19], where the population is assumed to be homogeneously connected. The individuals are classified into one of three states: (i) ignorant, i.e., someone that does not hold the information; (ii) spreader, i.e., someone that knows the rumor and is willing to spread it; and (iii) stifler, i.e., someone who knows the rumor but does not spread it. To be more specific, the terminology *ignorant* is widely used in rumor models to indicate someone unaware of the news in the sense that she/he did not hear about the news, this means someone who has not yet been exposed to the rumor. In DK model, the transitions are based on contact between individuals. A transmission occurs when there is a contact between a spreader and an ignorant, while an annihilation occurs upon a contact between two individuals who are aware of the rumor. In the DK model, the contacts are considered undirected, and a contact between two spreaders would result in the generation of two stiflers. For mathematical convenience, in 1973, Maki and Thompson (MT)[20] slightly changed the annihilation mechanism by considering directed contacts. In other words, when a spreader contacts another spreader, just the individual that initiated the contact turns into a stifler. This small change allowed for a series of analytical results. Another major advance was its generalization to complex networks in 2004[21], where the authors used a mean-field approach to show that no phase transition is predicted.

Despite having different purposes, rumor and disease spreading models are mathematically similar. Goffman and Newill were the first to notice the analogy between the spreading of a disease and information dissemination[19,22]. Both processes are typically modeled by compartmental models, where the population is divided into mutually exclusive and exhaustive classes. Also, the spreading of disease and rumor processes are often modeled in the same way (this is the case of SIS, SIR, SIRS, MT, and DK models). However, the removal mechanism is usually different. In rumor models (DK and MT), this mechanism is driven by the contact between individuals aware of the information (spreaders and stiflers), while in the epidemic case, it is spontaneous. This mechanism was initially proposed in ref. [19] and was motivated by the hypothesis that an active spreader stops telling the rumor because she/he learns that it has lost its 'news value.' The authors called this mechanism the "reluctance to tell stale news." Conversely, if one considers that the removal would be only due to a 'forgetting' mechanism (spontaneous), this process would follow the same equations as a SIR epidemic spreading model.

Mean-field methods are the most common approach for the analysis of spreading processes. They have proved to be useful for understanding the behavior of spreading processes in many contexts, especially in complex networks[17,18,21,23–29]. However, from the mean-field framework, one would expect that the MT rumor model does not have a phase transition[21,25]. This behavior is very intriguing, especially if compared with the SIR (Susceptible-Infected-Recovered) epidemic spreading processes. At first glance, these two processes are similar as both have infinitely many absorbing states. However, the two models have very different annihilation mechanisms. In rumor models, this mechanism is driven by the contact between individuals aware of the information, while in the epidemic case, it is spontaneous. The epidemic model is very well characterized, with most dynamics converging exponentially to the absorbing state in the subcritical regime[30] and a well-defined phase transition[17,18,25–28,30–32]. Surprisingly, the same can not be said about rumor models in networks as, to the best of our knowledge, up to now, the results regarding the absence of a phase transition were not challenged nor formally proven. Recently, there have been several works about spreading of rumors investigating new compartmental models with practical applications such as, for example, fake news or other dissemination processes in online social networks[33–38]. However, these works are concerned with proposing more realistic models without focusing on the phase transition analysis of classical models such as the MT model.

Perhaps due to the similarities with the epidemic processes or the apparent absence of phase transitions, the MT model remained relatively underexplored. However, one must not forget that the mean-field employed in refs. [21,25] assumes that the states of the nodes are independent, which, as we will show, is not reasonable in rumor models. Indeed, mean-field approaches for the MT model in networks presented a relatively high error compared to Monte Carlo simulation[18]. To the best of our knowledge, just a few works tackled the study of phase transitions in the MT model. In ref. [39], the authors explored a k-regular ring with the insertion of additional edges. In this structure, they proved the existence of a transition between two regimes, one where the final number of stiflers is at most logarithmic with respect to the population size, and another where the final number of stiflers grows algebraically with the population size. Recently, a transition in the MT model was reported, and its nature was associated with the spatial pattern of connections[5]. More specifically, the transition was shown to depend on the number of interactions between subpopulations[5]. However, we observe that this transition is more general, where the structure determines some critical properties, e.g., the critical point, but not the existence of the transition. The transition exists regardless of the structure. We notice that for some structures the transition might be vanishing in the thermodynamic limit but exists in a finite setup.

Here, we critically revisit the results in the literature using non-mean field methods together with Monte Carlo simulations. First, in the section "Maki–Thompson model" ($\delta = 0$), we show that the MT model presents a phase transition, which is at odds with most classical results, impacting on more than 50 years of research. Furthermore, we find a very particular power-law behavior in its subcritical regime, where the survival time of a rumor diverges as the spreading rate decreases. The subcritical behavior explains why the mean-field approaches fail to predict the transition. To better understand this class of processes, we propose a minor modification in the standard model, changing the process from infinitely many absorbing states to a single absorbing state (the entirely ignorant population), which preserves the MT model's essence and allows additional analytical and simulation tools. It is important to emphasize that, although the introduction of a

spontaneous transition from stifler to ignorant affects the number of absorbing states in the model, this should not change its universality class. As discussed in refs. [40,41] for the contact process and the SIS model, respectively, the authors showed that the universality class seems to depend only on the locality of the interactions and not on whether there is a single or infinitely many absorbing states. This modification allows us to obtain a statistical characterization of our models with high precision, presented in sections "Modified rumor model: Critical behavior" and "Modified rumor model: Subcritical behavior". Aside from the numerical aspects, we also provided a theoretical explanation for our findings in the section "Modified rumor model: Critical behavior". Using asymptotic analysis, in the section "Modified rumor model: Asymptotic analysis", we were able to show that the first-order mean-field approaches cannot predict a phase transition. This limitation can be related to the non-monotonic behavior of the lifespan as a function of $\lambda$ present in the subcritical behavior. On the other hand, by using the properties of Poisson processes and tree-like approximations, we are able to characterize both the sub-critical and the critical regimes. We summarize our work and present our final remarks in the section "Discussion".

## Results

**Rumor models**. Let us first define our model. In alignment with the DK and MT models, here we also have the same set of states (ignorant, spreader, or stifler), which are modeled by associating to each node $i$ three Bernoulli random variables, $X_i$, $Y_i$, and $Z_i$. If node $i$ does not know the rumor, it is classified as an ignorant ($X_i = 1$, $Y_i = Z_i = 0$). If it knows the rumor and is spreading it, it is called a spreader ($Y_i = 1$, $X_i = Z_i = 0$). However, if it knows the rumor but does not spread it, it is classified as stifler ($Z_i = 1$, $X_i = Z_i = 0$). Note that $X_i + Y_i + Z_i = 1$, implying that, for a fixed node, only one variable will be one, while the others will be zero. The spreading evolves through the contact between nodes defined by an undirected network, which is codified by the adjacency matrix $\mathbf{A}$, whose entries $\mathbf{A}_{ij}$ are equal to one if there is an edge between nodes $i$ and $j$ and zero otherwise. Our process is defined in continuous-time as a collection of Poisson processes. If the contact is between a spreader and an ignorant, the second node will learn the rumor and become another spreader at rate $\lambda$. On the other hand, if the contact happens between a spreader and someone that already knows the rumor (spreader or stifler), then the spreader that initiated the contact will lose interest in the rumor, thus becoming a stifler at a rate $\alpha$. We also introduce a forgetting mechanism, where each stifler spontaneously becomes ignorant at a rate $\delta$. We assume that only stiflers can forget the rumor as spreaders are actively trying to transmit the information. Denoting the state of the node $i$ as $(X_i, Y_i, Z_i)_i$, the above described local rules are expressed as

$$
\begin{aligned}
(0,1,0)_i + (1,0,0)_j &\xrightarrow{\lambda} (0,1,0)_i + (0,1,0)_j, \\
(0,1,0)_i + (0,1,0)_j &\xrightarrow{\alpha} (0,1,0)_i + (0,0,1)_j, \\
(0,1,0)_i + (0,0,1)_j &\xrightarrow{\alpha} (0,0,1)_i + (0,0,1)_j, \\
(0,0,1)_i &\xrightarrow{\delta} (1,0,0)_i.
\end{aligned}
\tag{1}
$$

A graphical representation of these transitions is presented in Fig. 1. Please see the Supplementary Videos for a visual example of the temporal evolution of the Monte Carlo Simulations for the standard Maki–Thompson model near the critical point and in the supercritical regime.

We remark that, if $\delta = 0$, we recover the original MT model. Notice that $\delta$ can be interpreted either as a forgetting mechanism or, for evolving rumors as, individuals who do not know the

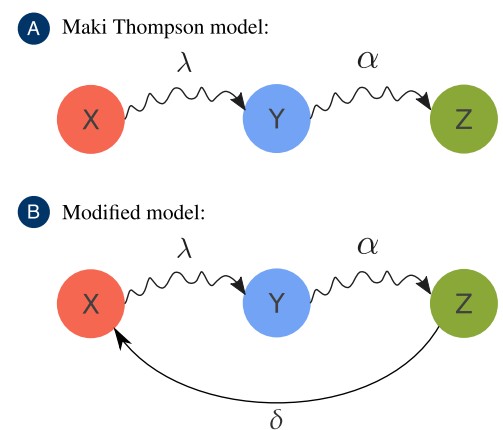

**Fig. 1 Graphical representation of the Maki Thompson and our modified model for each node.** In this figure, the curly arrows represent transitions driven by contact, while the normal arrow represents a spontaneous transition. To simplify the notation, the states of the nodes are represented in the center of the circles, which show the only Bernoulli random variable that is equal to one.

rumor's current state that is, someone that has not been informed since the rumor changed. In this case, we could simplify the process by treating an updated rumor as, in fact, a new one. Either way, the essence of the process is captured by the forgetting mechanism (with parameter $\delta$).

In summary, our model combines the "reluctance to tell stale news", the annihilation by contact, with the eventual forgetting of the rumor. The latter transition assumes that one can only forget a rumor if she/he is not spreading it. The removal mechanism is the central difference between our model and the SIRS model. Namely, in our modified model, the transition from spreaders to stiflers depends on a interaction-based removal mechanism, while in the disease case, this transition is spontaneous. Although these models share similar features, we observed that their differences are enough to generate a different behavior, which is especially clear in the subcritical regime. In this regime, the SIRS has an exponential decay, while our modified rumor model presents a power-law decay to the absorbing state.

From a Markovian point of view, the modified model has a single absorbing state, while the original MT model has infinitely many absorbing states. In other words, by including of the transition $Z_i \xrightarrow{\delta} X_i$ only the rumor-free state in which all the individuals are ignorant (i.e., $X_i = 1$ for all $i \in \{1, 2, \ldots N\}$) traps the dynamics. Note that, for the MT model any state in which $Y_i = 0$ for all $i \in \{1, 2, \ldots N\}$ is absorbing. In this case, as the system size increases, the number of absorbing states also increases. However, from a practical point of view, single absorbing state dynamics are easier to deal with as we can employ the quasi-stationary algorithm[42] (for more, see section "Monte Carlo simulations") and the lifespan method[43] (for more, see section "The lifespan method"), allowing for a more precise characterization of the process.

To analyze the model, we first need to define a phase transition and corresponding critical point in our context. As MT and SIR dynamics have infinitely many absorbing states, in this case, the critical point is defined as the parameter that separates two scaling regimes. Before this point, the final number of stiflers (or recovered in the SIR) when the process reaches an absorbing state does not scale with the system size and hence its fraction goes to zero in the thermodynamic limit. After the critical point, the number of stiflers (recovered) scales with the system size. On the other hand, for processes like SIS (Susceptible-Infected-

Susceptible) and our modified MT model with $\delta > 0$ where we have a single absorbing state (disease-free state), the critical point is defined as the point where above it we have an active state in the thermodynamic limit, and below it, the system goes to the absorbing state in the thermodynamic limit. Note that our model has infinitely many absorbing states if $\delta = 0$ and a single absorbing state if $\delta > 0$.

To characterize the MT model, we can simulate the process beginning with many different initial conditions and measure the final fraction of stiflers and the time necessary to reach the absorbing state. Note that the initial condition must correspond to a single spreader in an ignorant population to capture the transition. Thus, the initial spreader node changes in the different independent runs of our simulation. Formally, the quantities of interest can be defined as

$$\rho_Z = \left\langle \frac{\sum_{i=1}^{N} Z_i(t \to \infty)}{N} \right\rangle, \tag{2}$$

$$\tau = \left\langle \inf\left( t \,\middle|\, \sum_{i=1}^{N} Y_i(t) = 0 \right) \right\rangle, \tag{3}$$

where $\inf(\cdot)$ is the infimum and $\langle \cdot \rangle$ is the average among simulations. In this context, $\rho_Z$ measures how far the rumor spreads, while $\tau$ measures how fast it spreads. If $\delta > 0$, our modified model has a single absorbing state, and it thus does not make sense to characterize the behavior of the model based on the final state as in Eqs. (2) and (3). Instead, we measure how far the rumor spreads by estimating the density of spreaders, $\rho_Y$, in the metastable state using the quasi-stationary algorithm[42] (for more, see section "Monte Carlo simulations"). Further, the time to reach the absorbing state, $\tau$ (see Eq. (3)), diverges exponentially in the supercritical regime when $\delta > 0$. We instead estimate the lifespan of "finite" realizations, $\tau_f$, using the method proposed in ref. [43] (for more, see section "The lifespan method").

We remark that both the MT model and our modified version are represented by a continuous-time Markov chain with $3^N$ possible micro-states. Thus, it is theoretically possible to write the infinitesimal generator, leading to a linear system that solves these processes exactly. However, this is not possible in practice due to the prohibitive computational cost. Notice that this is the case of the SIR model or any three-state dynamics. Note also that the concept of phase transitions is only valid in the thermodynamic limit. However, as it is commonly done in complex systems[17,18,21,24,26], we also use the term phase transition in finite systems as we observe two different scaling behaviors, as mentioned above. Despite the fact that all our simulations are performed in a finite system, for this theoretical reason, most of our analysis implicitly assumes the thermodynamic limit. The exception here is section "Modified rumor model: Asymptotic analysis", which follows an individual-based approach.

**Maki–Thompson model ($\delta = 0$).** Our first main result is showing that the MT model has a phase transition. Figure 2 shows the average fraction of stiflers $\rho_Z$ as a function of $\lambda$ in the standard MT model with $\alpha = 1.0$ for random regular networks with different sizes. In this network, all nodes have the same number of neighbors $\langle k \rangle_k$, and the connections between them are made at random, avoiding both self and multiple connections. We observe that, as the system size increases, for small $\lambda$ the fraction of stiflers decreases with system size, whereas for large $\lambda$ it is larger than zero and weakly dependent on the system size (see Fig. 2 for an example). Note that, as we increase the system size, for a small $\lambda$ regime, i.e., $\lambda < \lambda_c$, where $\lambda_c$ is the critical point, the fraction of stiflers goes to zero. On the other hand, for the larger $\lambda$ regime, i.e., $\lambda > \lambda_c$, the curves converge to the value of the fraction in the

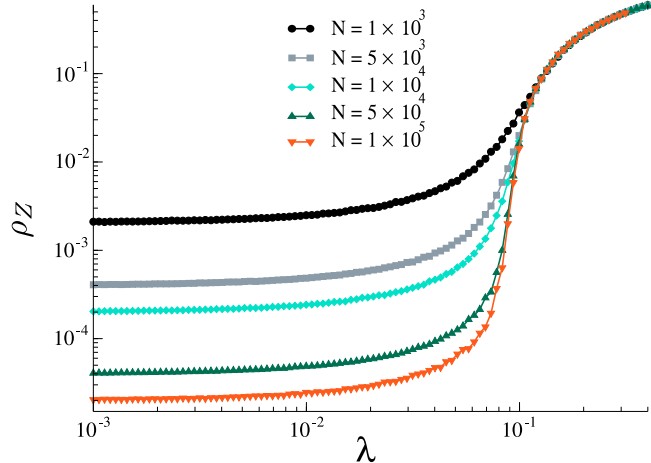

**Fig. 2 Phase diagram for the standard MT model.** Results for $\alpha = 1$ and different sizes on a random regular networks with $\langle k \rangle_k = 10$.

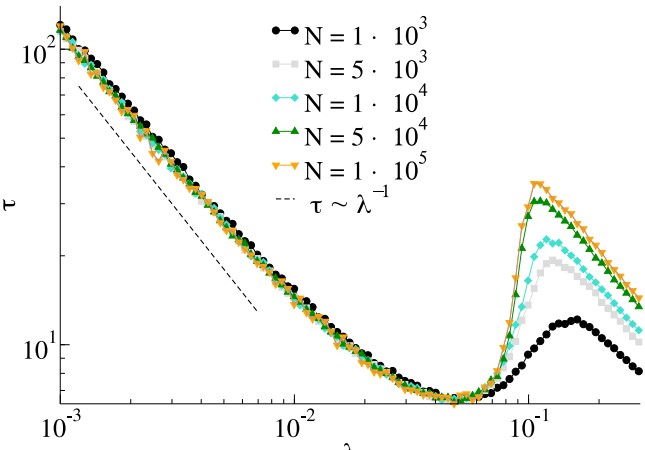

**Fig. 3 Time to reach the absorbing state for the standard MT model.** Results for $\alpha = 1$ and different sizes on random regular networks with $\langle k \rangle_k = 10$. The dashed line follows $\tau_f \sim \lambda^{-1}$.

thermodynamic limit. This behavior suggests a phase transition in the MT model as a function of $\lambda$. Complementary to the phase diagram, in Fig. 3, we show the corresponding lifespan. For a continuous phase transition, the peak of lifespan diverges in the thermodynamic limit[43,44]. By comparing Figs. 2 and 3 we can see that the curves for the order parameter meet at the point in which the lifespan diverges. This behavior is compatible with a continuous phase transition[43,44]. Moreover, we also observe an unexpected subcritical behavior, where the lifespan increases as $\lambda$ decreases. For the sake of comparison, in an SIS or SIR dynamics the lifespan would be an increasing function of $\lambda$ in the subcritical regime. Although these results are in striking contradiction with the mean-field approximations[17,18,21,25], we highlight that the mean-field approach neglects correlations between node states which are crucial in rumor models given the contact-driven stifling mechanism. Further, the first-order mean-field approximation is not accurate for rumor models[18].

**Modified rumor model: Critical behavior.** To better understand the transition in rumor models, we move to our modified model and concentrate our efforts on the $\alpha \gg \delta$ regime, which includes the MT model. In this case, in the transient period, we can neglect the forgetting transitions and assume that there are only two competing mechanisms, the spreading and the stifling. Thus, for

locally tree-like networks and near the absorbing state, we can estimate the expected number of newly informed nodes that result from the initial spread of the rumor to a node of degree $k$. Note that, at least one spreading event is required to reach the absorbing state from an initial state with one informed node due to the contact-driven stifling process. Formally, we have the following recurrent expression

$$q_k(i) = \begin{cases} \frac{(k-i)\lambda}{i\alpha+(k-i)\lambda}\left[ i\frac{(i+1)\alpha}{(i+1)\alpha+(k-i-1)\lambda} + q_k(i+1)\right] & \text{if } i < k \\ 0 & \text{otherwise,} \end{cases}$$

(4)

where $q_k(i)$ is an iterative expression that calculates the expected number of newly informed individuals in a process where the seed has $k$ neighbors and, among them $i$ are spreaders. This quantity should be iterated $k$ times and accounts for: (a) the probability of spreading the rumor to one of its $k - i$ available neighbors (ignorants), $\frac{(k-i)\lambda}{i\alpha+(k-i)\lambda}$; which is multiplied by (b) the probability that the process stops through a stifling event due to one of its $i$ spreader neighbors, $i\frac{(i+1)\alpha}{(i+1)\alpha+(k-i-1)\lambda}$, (c) or that the process continues, which is encoded in the term $q_k(i+1)$. In the case of the MT model on an infinite tree, the rumor propagation is exactly described by a branching process[45]. Here, we assume that each node's degree is sampled from a fixed degree distribution that is independent of the node and that there are no correlations between the degrees of neighboring nodes. By averaging over the degree distribution, the condition that establishes the transition between a phase where the rumor dies out with probability 1 and a phase where there is a non-zero probability of infinite propagation [45 Theorem 1, Chap. I] is

$$q(1) = \langle q_k(1) \rangle_k > 1,$$

(5)

where $\langle \cdot \rangle_k$ is the expectation on the degree distribution. We remark that Eq. (5) is not a closed expression, but can be numerically solved.

In other words, Eq. (4) estimates the expected number of newly informed individuals as a result from a single initial spreader event by weighting the probabilities that the rumor stops (accounted by item (a)) or that it continues the spreading (accounted by $q_k(i+1)$). Moreover, the transition can be obtained from the condition at which the expected number of newly informed nodes is larger than one, i.e., are more likely to spread the information than to stop the spreading. Note also that this approximation is based on the fact that only a single event occurs at a time, which is a property of continuous Markov chains.

In Fig. 4 we compare the solution of $q(1)$ with Monte Carlo critical point estimations for random regular networks with $\langle k \rangle_k = 10$ for both $\delta = 0$ and $\delta = 1$ (see section "Monte Carlo simulations" for the simulation details and the Supplementary Information, Sec. I, for the individual susceptibility curves and critical point estimations). In the same figure, we also present the naive estimation, considering a first-order approximation, given as $\lambda^* = \frac{\alpha}{(\langle k \rangle_k - 1)}$. Note that this expression accounts for the competition between the spreading processes, with the rate $2(\langle k \rangle_k - 1)$, and the annihilation, with the rate $2\alpha$. Moreover, $\lambda^*$ is the rate at which spreading and stifling have the same probability. So, from this approximation, one would expect that, if $\lambda > \lambda^*$, the process can spread to a fraction of the population, while if $\lambda < \lambda^*$ it should be constrained to a finite fraction of nodes. We observe that $q(1)$ seems to be a reasonable approximation as its precision increases with $\alpha$. On the other hand, $\lambda^*$ might be a reasonable approximation only for small enough $\alpha$. These solutions suggest that the process can not be reduced to a first-order approximation, and frustrated trials to get to the absorbing state are

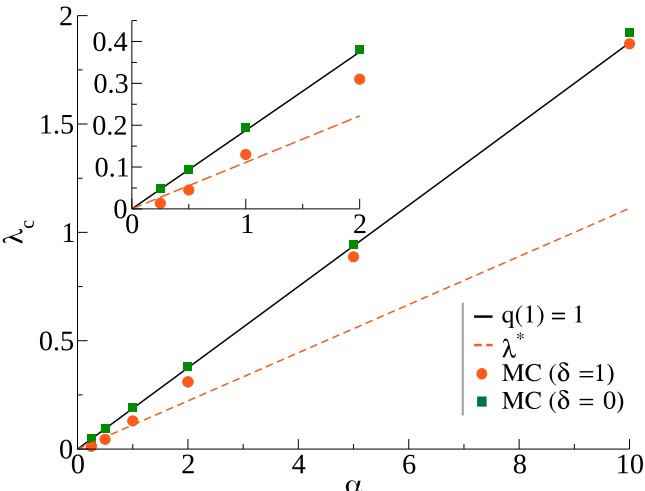

**Fig. 4 Comparison between analytical and Monte Carlo critical point estimations ($\lambda_c$).** Results for random regular networks with $\langle k \rangle_k = 10$ and $\delta = 1$ and $N = 10^6$. The continuous line expresses the value of $\lambda_c$ obtained as a solution of $q(1) = 1$, from Eq. (5). In contrast, the dashed line represents the naive approximation that accounts only for the probability that the next event is spreading or stifling. In the inset we present the comparison for the low $\alpha$ regime.

expected and should be accounted for. Here, a frustrated trial is defined as the scenario in which the system needs one event to be forced into the absorbing state, but a spreading event takes place instead, thus avoiding the absorbing state. A crucial difference between $q(1)$ and the mean-field approaches that fail to predict a phase transition is that $q(1)$ takes into account that the rate at which a spreader turns into a stifler increases with the number of spreader neighbors.

Typically, heterogeneity plays a major role on dynamical processes. To understand its effects in the phase transition, we analyzed the critical properties of uncorrelated random networks with a power-law degree distribution, $P(k) \sim k^{-\gamma}$, generated by the algorithm proposed by Catanzaro et al.[46], named uncorrelated configuration model. We remark that, here, the degree correlations can be measure by means of the conditional probability $P(k'|k)$ that a node with degree $k$ is connected with a vertex with degree $k'$. For uncorrelated networks, this probability can be estimated as the probability that any edge points to a vertex with degree $k'$, leading to $P_{unc}(k'|k) = k'P(k')/\langle k \rangle_k$. These networks are important from a numerical point of view because it is possible to test the behavior of dynamical systems whose theoretical solution is obtained only in the absence of correlations. Using the quasi-stationary method (see section "Monte Carlo simulations"), we estimated the critical point and focused our analysis on its behavior as a function of the network size. These results are summarized in Fig. 5 (please see the Supplementary Information, Sec. I, for the individual susceptibility curves and critical point estimations). Importantly, we observe that in power-law networks the threshold seems to vanish for $\gamma < 3.0$, and converges to a non-null value for $\gamma > 3.0$. This behavior is at odds with the behavior of the SIS model, in which the power-law networks have a vanishing critical point for any value of $\gamma$[18,47,48]. Conversely, it is similar to the SIRS (Susceptible-Infected-Recovered-Susceptible) model[49], contact process[50], the generalized SIS model with weighted infection rates[51], and also modified versions of the SIS model[32]. In all of these models, the phase transition can be associated with a collective phenomenon that involves the activation of the whole network, whereas the phase transition in the standard SIS model shows an unusual behavior related to mutual reinfection of hubs[43,52–54].

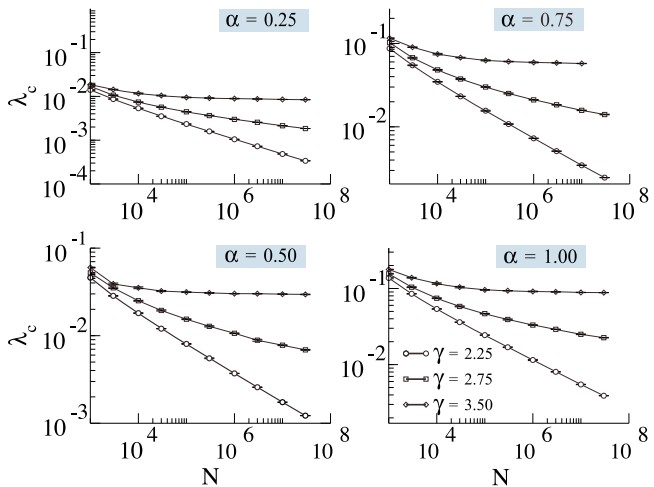

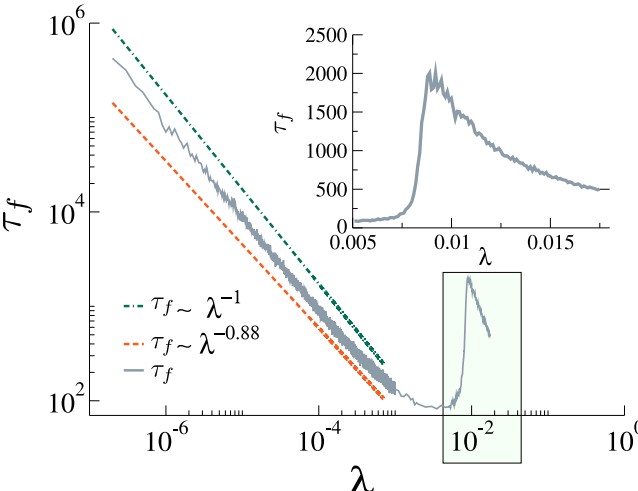

**Fig. 5 Critical point estimations of uncorrelated power-law networks.** We plot $\lambda_c$ as a function of $N$ and for different values of $\gamma$ and $\alpha$, considering $\delta = 1$.

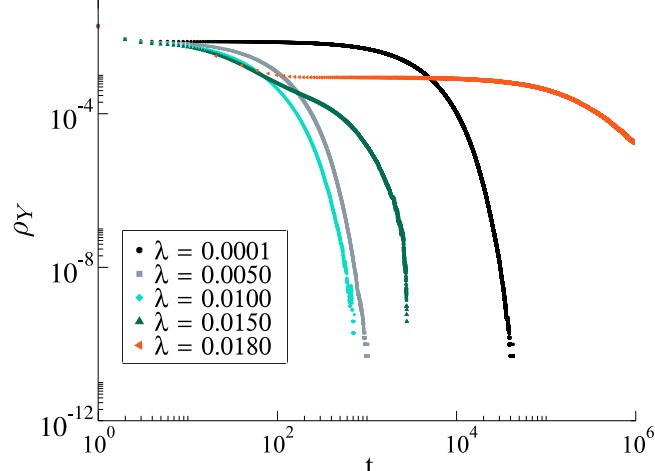

**Fig. 7 Temporal behavior of the density of spreaders for an uncorrelated power-law network.** Result for a network with $N = 10^6$, $\gamma = 2.25$, $\alpha = 0.5$ and $\delta = 1$ in the regime $\lambda \ll \delta$ for values of $\lambda$ near the critical point $\lambda_c \approx 0.015$.

**Fig. 6 Lifespan as a function of the spreading rate $\lambda$.** Results for $\delta = 1$, $\alpha = 0.5$ on an uncorrelated power-law network with $P(k) \sim k^{-\gamma}$ with $\gamma = 2.25$, $N = 10^6$. In the main panel, we show a wide range of $\lambda$, emphasizing the sub-critical behavior, while in the inset we show the peak that suggests a continuous phase transition. The blue curve (dot dashed line) follows $\tau_f \sim \lambda^{-1}$ and the orange curve (dashed line) follows $\tau_f \sim \lambda^{-0.88}$, obtained from a fitting of the lifespan obtained using Monte Carlo simulations (the gray curve).

**Modified rumor model: Subcritical behavior.** Another important result regards the subcritical regime, where the rumor can last for a "very long" time in the small $\lambda$ regime. To the best of our knowledge, this behavior remained unexplored until now. We briefly mentioned this phenomenon in the analysis of Fig. 3 for the MT model. For our modified model, these behaviors can be observed in Fig. 6, which shows the lifespan $\tau_f$ as a function of the spreading rate $\lambda$ for an uncorrelated power-law network with $P(k) \sim k^{-\gamma}$ with $\gamma = 2.25$ and $N = 10^6$. In the main figure, we show the subcritical regime, whose lifespan follows $\tau_f \sim \lambda^{-\eta}$, while in the inset, we show the peak for $\tau_f$, which suggests a continuous phase transition. To calculate the lifespan we used the method proposed by Boguñá and collaborators[43] (for more, see section "The lifespan method").

Complementarily, Fig. 7 shows the temporal behavior of the density of spreaders $\rho$, where we can see that the time to reach the

absorbing state is larger for $\lambda = 0.0001$ than for $\lambda = 0.01$, at odds with exponential decay. Phenomenologically, the transition from the spreader to the stifler depends on at least two individuals who are not ignorant. In the $\lambda \ll \delta$ regime, the forgetting mechanism reduces the number of stiflers faster than the creation of new spreaders, thus increasing the number of necessary events to reach the absorbing state.

The subcritical regime can be characterized in terms of the time to reach the absorbing state, $T_{\text{abs}}$. We remark that $\tau$ in Eq. (3) should converge to $T_{\text{abs}}$. This quantity can be approximated under the assumption that there is a single spreader and the population is locally tree-like, where nodes have $\langle k \rangle_k$ neighbors. In this case, assuming that node $i$ is a spreader, the shortest sequence of events that will drive the process to the absorbing state consists of: (1) node $i$ spreads the information to node $j$, (2) either $i$ or $j$ turn into stifler, (3) the other node turns into a stifler, (4) either $i$ or $j$ turn into ignorant, and (5) the other node turns into an ignorant. Since our process is a continuous Markov chain, only one event occurs at a time, the respective probabilities $P$ and expected times $\langle \tau \rangle$ of these events are

$$P_1 = 1 \qquad \langle \tau_1 \rangle = \frac{1}{\langle k \rangle_k \lambda} \qquad (6)$$

$$P_2 = \frac{2\alpha}{2\alpha + 2\lambda(\langle k \rangle_k - 1)} \quad \langle \tau_2 \rangle = \frac{1}{2\alpha + 2\lambda(\langle k \rangle_k - 1)} \qquad (7)$$

$$P_3 = \frac{\alpha}{\alpha + \lambda(\langle k \rangle_k - 1) + \delta} \quad \langle \tau_3 \rangle = \frac{1}{\alpha + \lambda(\langle k \rangle_k - 1)} \qquad (8)$$

$$P_4 = P_5 = 1 \qquad 2\langle \tau_4 \rangle = \langle \tau_5 \rangle = \frac{1}{\delta}. \qquad (9)$$

Therefore, the probability and the expected time for reaching the absorbing state through this chain are

$$P_{\text{abs}} = \frac{\alpha^2}{(\alpha + (\langle k \rangle_k - 1)\lambda)(\alpha + \delta + (\langle k \rangle_k - 1)\lambda)} \qquad (10)$$

$$\langle \tau_{1 \to 5} \rangle = \frac{1}{\langle k \rangle_k \lambda} + \frac{3}{2\delta} + \frac{3}{2\alpha + 2\lambda(\langle k \rangle_k - 1)}. \qquad (11)$$

From these quantities, we can approximate the average time necessary to reach the absorbing state, $\langle T_{\text{abs}} \rangle$. In the $\lambda \ll \delta$ and $\lambda \ll \alpha$ regime, $\langle \tau_1 \rangle \gg \langle \tau_\ell \rangle$, for $\ell = 2, 3, 4, 5$. Thus, the time spent

in the frustrated trials can be approximated by $\langle \tau_1 \rangle$. Here, we call as frustrated trials any sub-chain of events that does not lead to the absorbing state. So, $\langle T_{\text{abs}} \rangle$ can be approximated by counting the number of times the process fails to reach the absorbing state plus the time it succeed, which is given by

$$T = \sum_{i=1}^{\infty} i \langle \tau_1 \rangle (1 - P_{\text{abs}})^i + P_{\text{abs}} \langle \tau_{1 \to 5} \rangle \approx \langle T_{\text{abs}} \rangle, \quad (12)$$

which solves as

$$T = \frac{(\alpha + (\langle k \rangle_k - 1)\lambda)(\alpha + \delta + (\langle k \rangle_k - 1)\lambda)\left(((\langle k \rangle_k - 1)\lambda(2\alpha + \delta) + \alpha\delta + (\langle k \rangle_k - 1)^2\lambda^2\right)}{\alpha^4 \langle k \rangle_k \lambda}$$
$$+ \frac{\alpha^2 \left( \frac{1}{\alpha + \delta + (\langle k \rangle_k - 1)\lambda} + \frac{1}{2\alpha + 2(\langle k \rangle_k - 1)\lambda} + \frac{3}{2\delta} + \frac{1}{\langle k \rangle_k \lambda} \right)}{(\alpha + (\langle k \rangle_k - 1)\lambda)(\alpha + \delta + (\langle k \rangle_k - 1)\lambda)}, \quad (13)$$

and for the regime $\lambda \ll 1$ and $\lambda \ll \alpha, \delta$, follows

$$T^* = \left( \frac{(\alpha + \delta)\delta}{\langle k \rangle_k \alpha^2} + \frac{\alpha}{\langle k \rangle_k (\alpha + \delta)} \right) \lambda^{-1}. \quad (14)$$

Note that Eq. (13) is not defined for the standard MT model, i.e., $\delta = 0$. In the regime $\lambda \ll \alpha$, we expect that $T \sim \lambda^{-1}$ and only two nodes learn the rumor. It is noteworthy that, in the subcritical regime, the time to reach the absorbing state is dominated by the time the system has to wait before a spreading event happens. In order to reach the absorbing state, whenever we have a single spreader, it first needs to inform a neighbor and, only then, the process will be allowed to reach the absorbing state. Notice that, after the spreading event, the stifling events are much faster than the spreading ones. As the spreading events are the slowest ones in the subcritical regime, the rate of these processes dominates Eq. (13), as shown in Eq. (14). Also, frustrated trials to reach the absorbing state are possible. However, in this case, we are only repeating the above-described process. Figure 8 shows $T$ and $T^*$ as a function of $\lambda$, where the subcritical behavior is dominated by the time spent on the spreading events. Although our assumptions do not cover heterogeneous networks, $T$ has a similar behavior as the one observed in Fig. 6 for a power-law network, suggesting that our assumptions are reasonable and might be applicable in different substrates.

Note that in our modified model, $\delta > 0$, the rumor might still wander around in the network before die-out. For every frustrated attempt to reach the absorbing state, the spreader

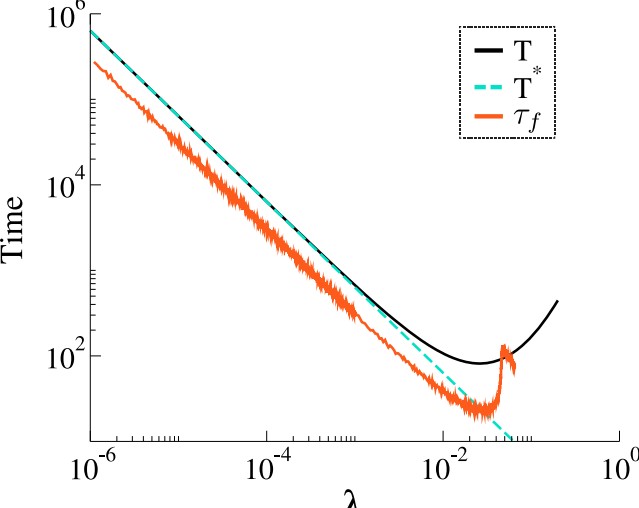

**Fig. 8 Approximations $T$ and $T^*$ for the time to reach the absorbing state as a function of $\lambda$.** Results for $\delta = 1.0$ and $\alpha = 0.5$. The orange curve is the result of Monte Carlo simulations in a random regular network with the same parameters, $\langle k \rangle_k = 10$ and $N = 10^5$.

individual might be a different one. In this way, although the rumor is fated to disappear, during the transient period, many different individuals might be informed about the rumor before it dies out. This effect is a consequence of the "reluctance to tell stale news" (the annihilation by contact) with the forgetting mechanism. The first mechanism imposes that at least two spreaders are necessary to reach the absorbing state, while the second mechanism enforces a single absorbing state.

**Modified rumor model: Asymptotic analysis.** From the analytical viewpoint, the analysis of arbitrary networks is much more difficult, and even the proof of the existence of a critical point is very hard or maybe impossible in a first-order approximation. This argument might be of interest to our community as it points to interesting future directions. In a first-order approximation we assume that the probabilities are independent. Denoting $\langle X_i \rangle$, $\langle Y_i \rangle$ and $\langle Z_i \rangle$ by $x_i$, $y_i$, and $z_i$, respectively, we have

$$\begin{cases} \frac{dx_i}{dt} = & \delta z_i - \lambda \sum_{k=1}^{N} \mathbf{A}_{ki} x_i y_k \\ \frac{dy_i}{dt} = & \lambda \sum_{k=1}^{N} \mathbf{A}_{ki} x_i y_k - \alpha \sum_{k=1}^{N} \mathbf{A}_{ki} y_i (y_k + z_k) \\ \frac{dz_i}{dt} = & -\delta z_i + \alpha \sum_{k=1}^{N} \mathbf{A}_{ki} y_i (y_k + z_k). \end{cases} \quad (15)$$

Here we follow an asymptotic analysis, considering that $y_i = y_i^{(1)} \epsilon^c + O(\epsilon^{2c})$, $z_i = z_i^{(1)} \epsilon^k + O(\epsilon^{2k})$ and $x_i \in O(1)$, where $\epsilon \ll 1$. Moreover, we have to consider a scaling of the parameters as $\lambda = \tilde{\lambda} \epsilon^m$ and $\alpha = \tilde{\alpha} \epsilon^n$ to be able to balance the equations. Thus, from Eq. (15) in the steady-state and neglecting the higher-order terms, we have

$$\begin{cases} \delta z_i^{(1)} \epsilon^k & = \tilde{\lambda} \epsilon^m \sum_{k=1}^{N} \mathbf{A}_{ki} x_i^{(1)} y_k^{(1)} \epsilon^c \\ \delta z_i^{(1)} \epsilon^k & = \tilde{\alpha} \epsilon^n \sum_{k=1}^{N} \mathbf{A}_{ki} y_i^{(1)} \epsilon^c \left( y_k^{(1)} \epsilon^c + z_k^{(1)} \epsilon^k \right). \end{cases} \quad (16)$$

This relation imposes that $k = n + c + \min(c, k)$ and $k = m + c$, establishing three different possible regimes: (i) $0 < c < k$, where $c = m - n$ and $k = 2m - n$, (ii) $c = k$, where $m = 0$ and $n = -k$, and (iii) $0 < k < c$, where $c = -n$ and $k = m - n$. Without loss of generality, we also assume $\delta = 1$. In the first regime, we have that $y_i^{(1)} = \frac{\tilde{\lambda}}{\tilde{\alpha}}$ and hence $y_i \sim \frac{\lambda}{\alpha}$. For the second and third regimes we have to assume a regular network (i.e., all nodes have the same degree) to obtain a closed-form solution, resulting in $y_i \sim \frac{\lambda}{\alpha(\lambda \Lambda_{\max} + 1)}$ for the second, and $y_i \sim \frac{1}{\alpha \Lambda_{\max}}$ for the third regime, where $\Lambda_{\max}$ is the leading eigenvalue of the adjacency matrix. We remark that $\Lambda_{\max} = k$ in regular networks. For the details and derivations of this analysis, see the "Methods" section. A similar asymptotic analysis was made for the SIRS model in the Supplementary Information, Sec. II, for the sake of comparison.

Note that, we can go from one regime to another by controlling the parameters $\lambda$ and $\alpha$. However, regardless of the regimes, $y_i$ is always positive and larger than zero for any positive non-null rates in the three possible regimes, thus implying that the phase transition is not captured in the first-order approximation. Nonetheless, the transition was observed in our numerical experiments. Thus, we can conclude that the assumptions on the first-order mean-field approach are not enough to capture the essence of rumor models and correlations should be included.

## Discussion
We studied both the standard MT model and a modified version with an additional transition from stifler to ignorant. This simple modification completely changes the thermodynamic behavior of the model. In the MT model, we have infinitely many absorbing states, while in our modified version, we have a single absorbing state (the exception is $\delta = 0$, in which we recover the MT model).

First, we show that the MT model has a phase transition in its dynamic behavior as a function of the spreading rate. In other words, for a one seed initial condition and controlling only the spreading parameter $\lambda$, there are two scaling regimes, one where the fraction of stiflers goes to zero in the thermodynamic limit and a regime where it scales with the population size. This contradicts the commonly accepted result that such a transition is absent in the MT model, which was based on the behavior of first-order mean-field approximations. In our modified model with $\delta > 0$, we have a single absorbing state, and the transition can be found between the rumor-free state and an active state. We provided an expression for locally-tree-like homogeneous networks that allow us to estimate the critical point, covering both the MT and modified models. Crucially, the tree-based (branching process) approximation, Eq. (4), explicitly accounts for local feedback effects that are ignored by the first-order mean-field approximations that fail to capture the transition. The key feature we capture in this approximation is that spreading events introduce a feedback loop, as they increase the probability that the initial spreading node is stifled. In the mean-field calculation, this effect is averaged out, which leads to underestimating the local rate of stifling. We characterized the phase transition both in random regular and power-law networks, showing that for $\gamma < 3.0$ the critical point seems to vanish in the thermodynamic limit, while for $\gamma \geq 3$ it converges to a non-null value. These findings seem to be robust against variations in $\alpha$. However, for fixed network size, as we increase $\alpha$, the critical point also increases. More interestingly, we observed a particular subcritical regime, whose lifespan follows $\tau^{-\eta}$ in the $\lambda \ll \delta$ regime, which we studied both analytically and numerically. Phenomenologically, the annihilation mechanism depends on the contacts and, if $\lambda$ is very small, the rumor needs a very long time to reach an absorbing state. Note that, if $\delta > 0$, the rumor might still wander around in the network before die-out. From a practical viewpoint, this means that, in order to contain the rumor spreading, people need to be aware of it. As our results might impact more than 50 years of research, we hope that our findings will motivate further research to better characterize the subcritical regime and phase transition in arbitrary networks analytically, including an accurate analysis of the universality class of such models, and experiments that might validate the peculiar dynamics of rumor spreading implied by our findings in real-world systems. We identified two main future research lines that might give an additional understanding of rumor dynamics. First, we believe an in-depth analysis of the universality class of rumor models, as well as understanding its relationship with dynamical percolation[55], might provide some additional mechanistic explanations for the observed phase transition. Second, we believe that the Geometric Singular Perturbation Theory, as applied for the SIRS[56], might provide additional analytical insights, especially in the lower $\lambda$ regime.

## Methods

**Monte Carlo simulations.** In this section, we will focus on the computational viewpoint to model the rumor process described in the main text. Denoting by $N_y$ the number of spreaders, $M_y$ the number of edges emanating from spreaders, and $N_z$ the number of stiflers. We implemented our model as follows. At each step, with probability $\delta N_z/[\delta N_z + (\lambda + \alpha)M_y]$ one stifler, chosen at random, forgets the rumor and becomes an ignorant again. With probability $(\lambda + \alpha)M_y/[\delta N_z + (\lambda + \alpha)M_y]$ one contact is made. Algorithmically, this contact is implemented in two steps: (i) A spreader vertex $i$ is selected with probability proportional to its degree, next (ii) a nearest neighbor of $i$, here denoted as $j$, is selected uniformly. If $j$ is an ignorant, with probability $\lambda/(\lambda + \alpha)$, $j$ learns the rumor and becomes a spreader. If $j$ is another spreader or a stifler, with probability $\alpha/(\lambda + \alpha)$ it becomes a stifler. If none of these conditions are satisfied, nothing happens. Next, time is incremented by $dt = 1/[\delta N_z + (\lambda + \alpha)M_y]$. With this scheme, we have two different algorithms, one for $\delta > 0$, where we have a single absorbing state (rumor-free), and another for

$\delta = 0$, where we have a strong dependency on the initial condition and many absorbing states.

For the $\delta > 0$ case, the simulations were performed using the quasi-stationary method[42], which is one of the most robust approaches to overcome the stationary simulations' difficulties of finite systems with absorbing states. In this method, the quasi-stationary probability, named $\bar{P}_n$, is defined as the probability that the system has $n$ spreader vertices in the quasi-stationary regime. Every time the process tries to visit an absorbing state in the quasi-stationary simulation, this state is substituted by an active configuration previously visited during the simulation. These active configurations are stored in a list, which is constantly updated and works as a new initial condition. This approach is completely equivalent to the standard quasi-stationary method where averages are performed only over samples that did not visit the absorbing state[57]. The density of spreader nodes is derived from $\bar{P}_n$ as $\rho_Y = \frac{1}{N}\sum_n \bar{P}_n$[42]. The lifespan is also related with the quasi-stationary density as $\tau = 1/\bar{P}_1$[42]. However, we used the method of ref. [43] to obtain a more robust estimate (see section "The lifespan method"). In our experiments, we let the simulations run during a relaxation time $t_r = 10^7$ time steps, and after the relaxation period, we computed the averages $\bar{P}_n$ over $t_{av} = 10^7$ time units. The critical point can be estimated using the modified susceptibility[58],

$$\chi = \frac{\langle n^2 \rangle - \langle n \rangle^2}{\langle n \rangle} = N\left(\frac{\langle (\rho_Y)^2 \rangle - \langle \rho_Y \rangle^2}{\langle \rho_Y \rangle}\right), \tag{17}$$

where $n$ is the number of spreaders, and $\rho_Y$ is the quasi-stationary density. As argued in[18,58,59], the susceptibility presents a peak at the phase transition in finite systems. The parameters corresponding to the maximum value of the susceptibility will coincide with the critical threshold for sufficiently large systems.

For the $\delta = 0$ case, the algorithm consists of running the simulation, beginning with a single randomly placed seed and calculating the final fraction of stiflers. To have a better estimation, we run in the range of 500 up to $5 \times 10^5$ independent simulations, guaranteeing that $\chi$ does not vary more than $10^{-3}$ if compared with the result before the last batch of 500 simulations. We highlight that, although in this case, we are not interested in the susceptibility, as it depends on the second moment of the distribution of stiflers, this will guarantee a reasonable sampling for the first moment and its peak was also a reasonable indicator of the transition as well. We also remark that, in our experiments, the susceptibility peak coincides with the lifespan peak calculated using the method proposed in ref. [43].

**The lifespan method.** The lifespan method, proposed by Boguñá et al.[43] can be used to study critical properties of generic absorbing-state phase transitions. In this method, the spreading simulation starting from a single seed. Each realization is characterized by its coverage $C$—the fraction of different vertices which have been a spreader at least once during the simulation—and its lifespan $\tau$.

In the thermodynamic limit, there are two possibilities: realizations can be considered *finite* or *infinite*, depending on whether they are below or above the critical point. Endemic realizations are characterized by an infinite lifespan and a network-wide coverage, while finite realizations have a finite lifespan and a coverage vanishingly small in the limit of diverging size.

However, in finite systems any realization can reach the absorbing state sooner or later due to dynamical fluctuations. Thus, we assume that a realization is active whenever its coverage reaches a predefined threshold value $C_t$. The method is robust with respect to the coverage threshold $C_t$[44], which can be considered, for example, equal to $C_t = 50\%$ of the network size. Realizations ending before value $C = C_t$ is reached are considered to be finite.

The role of the order parameter is played by the probability Prob$(\bar{\rho}^Y \geq Ct, \lambda, N)$ that a realization is long-term, it means, the probability that a run reaches the predefined coverage Ct (i.e., it is effectively endemic), while the role of susceptibility is played by the average lifespan of finite realizations, $\tau_f$. For $\lambda$ close to but below the critical point, all realizations usually have a finite and very short lifespan $\tau$. As $\lambda$ grows the average duration of finite realizations increases, diverging at $\lambda_c$. However, for $\lambda > \lambda_c$, the realizations that remain finite have necessarily a short lifespan since the probability of samples reaching the active phase increases. So, in this range, $\tau_f$ also decreases as $\lambda$ increases. Thus, $\tau_f$ exhibits a peak depending on $N$ and converging to $\lambda_c$ in the thermodynamic limit.

**Asymptotic analysis.** Equation 15 in the main text is the first-order approximation of our rumor model. As the annihilation mechanism always depends on second-order terms, in Eq. (15) by the products $y_i y_k$ and $y_i z_k$, simply neglecting higher-order terms will always exclude the stifling processes (the ones associated with $\alpha$). In practice, this limitation can be overcome by considering that the scaling of the parameters be related with the scaling of the nodal probabilities. This approach will allow us to understand the behavior of our model. From Eq. (15) and considering that $y_i = y_i^{(1)}\epsilon^c + O(\epsilon^{2c})$, $z_i = z_i^{(1)}\epsilon^k + O(\epsilon^{2k})$ and $x_i \in O(1)$, where $\epsilon \ll 1$. Moreover, consider the scaling of the parameters as $\lambda = \tilde{\lambda}\epsilon^m$ and $\alpha = \tilde{\alpha}\epsilon^n$ to be able to balance the equations. Thus, from Eq. (15) in the steady-state and neglecting the higher-order terms, we arrive at Eq. (16). Without loss of generality, we also assume $\delta = 1$. Thus, the relations in Eq. (16) impose that $k = n + c + \min(c, k)$ and $k = m + c$, which establish three different possible regimes: (i) $0 < c < k$, where $c = m - n$ and $k = 2m - n$, (ii) $c = k$, where $m = 0$ and $n = -k$, and

(iii) $0 < k < c$, where $c = -n$ and $k = m - n$. In the following, we evaluate each case individually.

In the first regime, $0 < c < k$, we have that

$$z_i^{(1)} = \tilde{\lambda} \sum_j \mathbf{A}_{ij} y_j^{(1)} \tag{18}$$

$$z_i^{(1)} = \tilde{\alpha} \sum_j \mathbf{A}_{ij} y_i^{(1)} y_j^{(1)}. \tag{19}$$

Defining $\boldsymbol{y} = \left[ y_1^{(1)}, y_2^{(1)}, \dots, y_N^{(1)} \right]^T$ and plugging Eq. (18) into Eq. (19) we have

$$\tilde{\lambda} \mathbf{A} \boldsymbol{y} = \tilde{\alpha} \boldsymbol{y} \circ (\mathbf{A} \boldsymbol{y}), \tag{20}$$

where "∘" denotes the Hadamard (i.e., element-wise) product. Consequently, this relation yields

$$y_i^{(1)} = \frac{\tilde{\lambda}}{\tilde{\alpha}}. \tag{21}$$

Substituting the scaling relations, this is equivalent to

$$y_i = \frac{\lambda}{\alpha} \epsilon^{n-m+c} + O(\epsilon^{2c}) = \frac{\lambda}{\alpha} + O(\epsilon^{2c}), \tag{22}$$

which shows that in this regime, at leading order, the probability of each node being a spreader depends only on $\lambda$ and $\alpha$. Moreover, for positive values of $\lambda$ and $\alpha$, $y_i$ will be positive and does not depend on the network.

Next, for the second case, considering only the dominating terms of Eq. (16) in the regime $c = k$, we have

$$z_i^{(1)} = \tilde{\lambda} \sum_j \mathbf{A}_{ij} y_j^{(1)}, \tag{23}$$

$$z_i^{(1)} = \tilde{\alpha} \sum_j \mathbf{A}_{ij} y_i^{(1)} \left( y_j^{(1)} + z_j^{(1)} \right). \tag{24}$$

Similar to the previous derivation, plugging Eq. (23) into Eq. (24), we obtain

$$\tilde{\lambda} \mathbf{A} \boldsymbol{y} = \tilde{\alpha} \boldsymbol{y} \circ (\mathbf{A} \boldsymbol{y} + \tilde{\lambda} \mathbf{A}^2 \boldsymbol{y}). \tag{25}$$

We do not have a closed-form solution for Eq. (25) for general networks. However, if we assume that $\mathbf{A}$ represents a regular network and $\boldsymbol{y}$ is a constant vector, we have $\mathbf{A} \boldsymbol{y} = \Lambda_{max} \boldsymbol{y}$, where $\Lambda_{max}$ is the leading eigenvalue of $\mathbf{A}$. Hence,

$$\tilde{\lambda} \Lambda_{max} y_i^{(1)} = \tilde{\alpha} y_i^{(1)} (\Lambda_{max} + \tilde{\lambda} \Lambda_{max}^2) y_i^{(1)}, \tag{26}$$

which yields

$$y_i^{(1)} = \frac{\tilde{\lambda}}{\tilde{\alpha} \left( \tilde{\lambda} \Lambda_{max} + 1 \right)} \tag{27}$$

and

$$z_i^{(1)} = \frac{\tilde{\lambda}^2 \Lambda_{max}}{\tilde{\alpha} \left( \tilde{\lambda} \Lambda_{max} + 1 \right)}. \tag{28}$$

Substituting the scaling relations as before, we obtain

$$y_i = \frac{\lambda}{\alpha(\lambda \Lambda_{max} + 1)} + O(\epsilon^{2c}), \tag{29}$$

$$z_i = \frac{\lambda^2 \Lambda_{max}}{\alpha(\lambda \Lambda_{max} + 1)} + O(\epsilon^{2c}). \tag{30}$$

As in the previous regime, $y_i > 0$ for positive values of $\lambda$ and $\alpha$. Note that in the regime $c = k$ there is a dependency on the network structure, here codified in the leading eigenvalue. However, such a dependency does not define a critical point.

Finally, considering the third regime, $0 < k < c$, we consider the dominating terms of Eq. (16) for the regime $0 < k < c$. Thus, we have

$$z_i^{(1)} = \tilde{\lambda} \sum_j \mathbf{A}_{ij} y_j^{(1)}, \tag{31}$$

$$z_i^{(1)} = \tilde{\alpha} \sum_j \mathbf{A}_{ij} y_i^{(1)} z_j^{(1)}. \tag{32}$$

Following the same approach as before, we have

$$\tilde{\lambda} \mathbf{A} \boldsymbol{y} = \tilde{\alpha} \boldsymbol{y} \circ (\mathbf{A}^2 \boldsymbol{y}). \tag{33}$$

Next, assuming that $\mathbf{A}$ represents a regular network and using the same argument as in the previous section, we have

$$y_i = \frac{1}{\alpha \Lambda_{max}} + O(\epsilon^{2c}). \tag{34}$$

Again we observe that $y_i > 0$. However, in this regime, this condition depends only on $\alpha$ and the network structure through $\Lambda_{max}$.

**Reporting summary**. Further information on research design is available in the Nature Research Reporting Summary linked to this article.

## Data availability
All the data used in our manuscript is artificially generated by computational simulations whose methods are explained in the text. The data is available from the corresponding author upon request.

## Code availability
The algorithms used in our experiments are described in the "Methods" section. Custom code implemented in C/C++ and Fortran used in this study is available from the corresponding author upon request.

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

## Acknowledgements

Angélica S. Mata acknowledges FAPEMIG (Grant No. APQ-01294-21) and CNPq (Grant No. 423185/2018-7). Guilherme F. de Arruda, Yamir Moreno, and Lucas Jeub acknowledge support from Intesa Sanpaolo Innovation Center. Yamir Moreno acknowledges partial support from the Government of Aragón and FEDER funds, Spain through grant ER36-20R to FENOL, and by MCIN/AEI and FEDER funds (grant PID2020-115800GB-I00). Francisco Rodrigues acknowledges CNPq (grant 309266/2019-0) and FAPESP (grant 19/23293-0) for the financial support given for this research. Research carried out using the computational resources of the Center for Mathematical Sciences Applied to Industry (CeMEAI) funded by FAPESP (grant 2013/07375-0). The funders had no role in study design, data collection, and analysis, decision to publish, or preparation of the manuscript.

## Author contributions

G.F.A., A.S.M., F.A.R., and Y.M. conceived and designed the study; G.F.A., L.G.S.J., and A.S.M. performed the calculations; G.F.A., L.G.S.J., A.S.M., F.A.R., and Y.M. analyzed, discussed the results, wrote the paper and contributed to the revision of the final manuscript.

## Competing interests

The authors declare no competing interests.
