## [Peer Review File · Nature Communications]

From subcritical behavior to a correlation-induced transition in rumor modelsREVIEWER COMMENTS

Reviewer #1 (Remarks to the Author):

Report on Nature Communications 323671

From subcritical behavior to a correlation-induced transition in rumor models

by Ferraz de Arruda et al.

The authors discuss several models of rumor spreading, in particular the MT model and a modification thereof proposed in the present work. They present results of Monte Carlo simulations and some analytic approximations showing that, contrary to currently held belief, the MT model on regular networks in fact exhibits a phase transition as the spreading parameter λ is varied. This conclusion may have significant implications for the understanding and possible response to rumor spreading. The manuscript, while quite interesting, has a number of defects, as discussed below. I would support publication of a suitably revised paper.

Models

The MT models bear a certain resemblance to epidemic models, the key difference being that while infection of a susceptible is autocatalytic in both cases, recovery is spontaneous in epidemics, but cooperative in the rumor model. It seems plausible that a spreader interacting with a stifler might yield another stifler. By contrast, the interaction of two spreaders converting one to a stifler is rather counterintuitive. This feature, which is essential for the model to exhibit interesting behavior, seems artificial and deserves some motivation on the part of the authors.

[If personal experience be any guide, some people are dedicated rumor spreaders, who only stop spreading rumor A when they become enchanted with some new rumor B, to whose spread they dedicate all their effort, until rumor C comes along... QAnon rumormongers hold conventions: Meeting up with fellow adherents doesn't seem produce stiflers. The MT models may in fact be better suited to describe the spread of a fashion: Fashions spread, at least in part, for their perceived novelty; someone who flaunts a given fashion may abandon it when, seeing others wearing the same style, their sense of belonging to a vanguard is destroyed.]

Incidentally, the term "ignorants" to describe those as yet unexposed to the rumor seems odd, almost offensive. Why not use "susceptibles" or "unexposed" to label this segment of the population?

Summing up, I believe the authors should provide a stronger motivation for the MT models and their relation to models of epidemics, to dynamic percolation [see P Grassberger, MATHEMATICAL BIOSCIENCES 63: 151-172 (1983)], fashions, opinions, and chain letters.

Absorbing States

Once the number of spreaders is zero, the rumor is fated to die. In the original MT model, as in an SIR epidemic, this end state is nonunique. Here, the authors introduce a (spontaneous) transition from stifler to ignorant (analogous to changing

from SIR to SIRS) to eliminate the multiplicity of the absorbing state. But this would appear to change the universality class, just as SIR (with no active steady state) belongs to dynamic percolation whereas SIRS or SIS (which do admit active steady states) belong to directed percolation. The original MT model is intrinsically a spreading process, whereas the modified MT model can maintain activity indefinitely. The authors' motivation for this alteration (that a unique absorbing configuration allows them to use quasistationary simulations) seems neither correct nor desirable, if it changes the universality class.

Network structure

The term "random regular network" does not appear to be standard in the literature, though a recent review article by one of the authors gives to understand that this is an Erdos-Renyi graph. In any case, a more precise definition and some details of the construction algorithm as needed. An illustration of a typical network, and of the progress of rumor spreading on it, would be highly desirable. (Videos of illustrative examples would also be helpful!)

Citations

The topic of rumor spreading on networks has been discussed in many recent studies. The authors need to enlarge and diversify their citations to the recent literature.

In summary, the authors have demonstrated that the MT model, in its original version, exhibits a phase transition. This result, which corrects a longstanding misconception based on a simple mean-field analysis, seems the most significant finding in the paper, and certainly merits publication. I would support acceptance of a manuscript revised to address the issues cited above.

Note: I'm attaching an annotated version of the paper.

Reviewer #3 (Remarks to the Author):

please see attached file

Review of “From subcritical behavior to correlation-induced transition in rumor models”

by Ferraz de Arruda and et al.

The authors present interesting observations about the rumor spreading model of Maki Thompson and their “delta” variation. The introduction is very nice; the Fig. 1 & 2 are crucial and new.

Their paper is publishable, in my opinion, after a serious rewriting. Below, I motivate this recommendation.

Even though the mathematical description of the model (Sec II.a) is poor, I believe that their model is correct: At each time t , each node is either Ignorant (susceptible) X , Spreader (infected) Y or Stifler (recovered) Z . Let me explain my confusion.

I believe that X, Y, Z are not Bernoulli random variable, but they describe the state of node i . Indeed, if X, Y, Z are Bernoulli random variables, what is the physical meaning of $X = 0, Y = 0$ and $Z = 0$ (only $X=1, Y = 1$ and $Z= 1$) is explained. Later on, $Y = 0$ is mentioned and infinitely many absorbing states (which is really confusing because a Markov process on a finite graph has rarely more than 2 absorbing states).

Their notation is slightly confusion, because the strength of continuous-time Markov processes lies in the fact that at any time, only one transition occurs. Here, I believe that, within each time step Δt , and with the authors’ definition of X_i, Y_i and Z_i , multiple values change for each transition. Nevertheless, I think that a precise description can be resolved. The corresponding Markov chain still consists of 3^N states (like in SIR, even though this is never mentioned explicitly).

The model is now explained in the thermodynamic limit (i.e. for infinitely large graphs), while reality and their simulations are finite. I think that it is better to explain the model for finite size and occasionally talk about the limit.

Suggestions:

a) the MT model is relatively unknown, while the epidemic theory is huge. Hence, I would draw the analogy between SIS, SIR and SIRS much better and clearly point (a) what is the same and (b) where the difference lies. Next to the “chemical” presentation in II. A, a figure of the transitions in node i may be useful.

b) Move the asymptotic analysis to the appendix or supplementary material. Why? First, eq. (10) are SIRS equations in first-order mean field. Only the last equation deviates from the many analyses of SIR (e.g. see the book of Kiss, Miller & Simon). The notation x, y, z used is due to Kermack-McKendrick. Second, the only point is to show that, as mentioned frequently in the paper, that a first-order mean-field analysis does not explain Fig. 1 & 2.

Specific small remarks to improve the paper:

- In the abstract, it says: ...“trusted or not”. I think that the authors mean “...trustworthy or not”.
- In the introduction, the authors states that “...we were able to show that the mean-field approaches cannot predict a phase transition.” The statement is only correct while considering first-order mean-field methods. Perhaps higher order mean-field approximation may capture this property.

- I do not understand the second part of this statement: "Notice that δ can be interpreted as either as a forgetting mechanism or, for evolving rumors, individuals who do not know the rumor's current state."
- Sec. II. B: "the fraction of stiflers ... independent of system size"? Strong statement and incorrect, I think. There is dependence, but, perhaps, weak.
Next sentence: "these results indicate a phase transition" (why?)
"peak of lifespan diverges...limit": add reference, as well as, in the sentence "this behavior is compatible with a second-order phase transition".
- The authors devote a lot of time/space to the subcritical regime, where λ is very small. In my opinion, the observed powerlaw behaviour is due to the process with rate α : Spreader nodes can only become stiflers while being in contact with a spreader or with another stifler. If the network is initiated with a single spreader node (which is the case here), then that node is only connected to ignorant nodes. Considering the limit of small "infection" rates λ , the spreader node must infect at least one neighbouring ignorant node before the "curing" process can take place. Thus, we need to wait for one λ -event to take place (which is slow, because λ is small) whereafter two α events take place (which are fast compared to λ) and then the process has converged to the steady state. I think the small λ regime is currently not well explained in the paper. Finally, I think that an analysis for small λ (see e.g. Geometric Singular Perturbation Theory, <https://arxiv.org/abs/2011.02169>) is possible, although it is not necessary here at all.
- In Sec II.B.3 the authors explain their asymptotic analysis but the statement "the proof of existence of a critical point is impossible in a first-order approximation" is too strong, even though I agree a proof will be hard.
- Sec. II. C.1: eq. (3) seems to hold in a k -array tree. A bit more explanation would be helpful. It seems that standard branching process theory is applied?
 - Explain or give a reference to "first-step approximation" (see e.g the books of Karlin & Taylor)
 - What are frustrated trials?
 - "At odds with power-law networks ... vanishing critical point for any ... γ ": I think that each network with an average degree that increases unboundedly with size N has a "SIS/SIR vanishing epidemic threshold" in the limit $N \rightarrow \infty$.
- Sec. III: "Crucially, this expression ... local correlations ...: I assume that "this expression" is eq. 3? Since eq. 3 is a recursion for the mean of a random variable in a branching process, I do not understand where "correlations" enter? A mean of a random variable does not incorporate correlations (i.e. joint probabilities). Also, "if $\delta > 0$, the rumor might still wander around ... before die-out" and the next sentence can be written more clearly to convey the message.

Minor issues:

- Spelling: E.g. "In this case, as the system size increases, the number of absorbing states also increases." (Note: the original text has a typo 'S' at the end)
- General: try to avoid "this and that", but be precise in what "this" mean. Each "this" implies that a reader must "substitute this" by the relevant quantity.
- Fig 1: legend for black curve show 1×10 , but it should probably be 1×10^3 (see fig. 2)

- Which method is used by the authors to generate random regular graphs? Also, we remark that the class of random regular graph is a rather small, special class. There are less random regular graphs than co-spectral graphs!
- Fig. 3: λ_c is not defined.
- Fig. 4: what is an “uncorrelated” power-law network?
- Fig 6: the horizontal time axis better uses 10^1 , 10^2 , 10^3 as time units
- Fig 6: the caption says $\lambda \gg \delta$, but I think it should be the reverse: $\lambda \ll \delta$

From subcritical behavior to a correlation-induced transition in rumor models
(NCOMMS-21-29679)
Reply to the Reviewers
(Dated: January 13, 2022)

DETAILED REPLY TO THE REPORT OF REVIEWER 1

Comment 1.1

The authors discuss several models of rumor spreading, in particular the MT model and a modification thereof proposed in the present work. They present results of Monte Carlo simulations and some analytic approximations showing that, contrary to currently held belief, the MT model on regular networks in fact exhibits a phase transition as the spreading parameter λ is varied. This conclusion may have significant implications for the understanding and possible response to rumor spreading. The manuscript, while quite interesting, has a number of defects, as discussed below. I would support publication of a suitably revised paper.

Reply 1.1.— We thank the reviewer for their thorough evaluation of our work. In this new version, we took into account all the reviewer’s comments, which indeed improved our manuscript. We hope that the revised version is now suitable for publication.

Comment 1.2

Models

The MT models bear a certain resemblance to epidemic models, the key difference being that while infection of a susceptible is autocatalytic in both cases, recovery is spontaneous in epidemics, but cooperative in the rumor model. It seems plausible that a spreader interacting with a stifter might yield another stifter. By contrast, the interaction of two spreaders converting one to a stifter is rather counterintuitive. This feature, which is essential for the model to exhibit interesting behavior, seems artificial and deserves some motivation on the part of the authors. [If personal experience be any guide, some people are dedicated rumor spreaders, who only stop spreading rumor A when they become enchanted with some new rumor B, to whose spread they dedicate all their effort, until rumor C comes along... QAnon rumormongers hold conventions: Meeting up with fellow adherents doesn’t seem produce stiflers. The MT models may in fact be better suited to describe the spread of a fashion: Fashions spread, at least in part, for their perceived novelty; someone who flaunts a given fashion may abandon it when, seeing others wearing the same style, their sense of belonging to a vanguard is destroyed.]

Reply 1.2.— The interaction between two spreaders generating a stifter was initially proposed by Daley and Kendall in their seminal work [1], and this interaction was maintained in the work by Maki and Thompson [2]. In both works, the motivation for such interaction is that if you contact someone that already knows the rumor, then you lose interest in spreading it because it might look like this is not “new” anymore. Note also that these works were published in 1964 and 1973, respectively, and were based in word-to-mouth rumors. So, in order to analyze the original model and its transition, we kept this transition in our model. However, we do agree that in modern works, this transition might be counterintuitive in some contexts. So, in the text, we better justify our choices.

Notice that the example provided by the reviewer is precisely one of the motivations for our modified model, in the case of “dedicated rumor spreaders, who only stop spreading rumor A when they become enchanted with some new rumor B, to whose spread they dedicate all their effort until rumor C comes along...”, we can interpret the rumor as an evolving rumor. Admittedly, this assumption implies a series of approximations regarding the rumor. However, the essence of the process is captured by the forgetting mechanism (with parameter δ). Finally, we hope that our model motivates other researchers to explore more complex models in these directions.

Moreover, the Maki-Thompson model inspired and constituted the theoretical basis for the so-called gossip protocol [3]. This protocol is widely used in the design of distributed systems. So, by better understanding the original MT model, we might also provide new insights for this application.

- [1] D. J. Daley and D. G. Kendall, Epidemics and rumours, *Nature* 204, 1118 (1964).
- [2] D. P. Maki and M. Thompson, *Mathematical models and applications* (Prentice-Hall Inc., Englewood Cliffs, N.J., 1973).
- [3] A. Demers, D. Greene, C. Hauser, W. Irish, J. Larson, S. Shenker, H. Sturgis, D. Swinehart, and D. Terry, in *Proceedings of the Sixth Annual ACM Symposium on Principles of Distributed Computing* (Association for Computing Machinery, New York, NY, USA, 1987), PODC '87, p. 1–12, ISBN 089791239X.

Action taken 1.2.— We better explained the origin and original motivations of the transitions in the text. Particularly, in Sec. II A we wrote the following explanation

In summary, our model combines the “reluctance to tell stale news”, the annihilation by contact, with the eventual forgetting of the rumor. The latter transition assumes that one can only forget a rumor if she/he is not spreading it. The removal mechanism is the central difference between our model and the SIRS model. Namely, in our modified model, the transition from spreaders to stiflers depends on a interaction-based removal mechanism, while in the disease case, this transition is spontaneous. Although these models share similar features, we observed that their differences are enough to generate a different behavior, which is especially clear in the subcritical regime. In this regime, the SIRS has an exponential decay, while our modified rumor model presents a power-law decay to the absorbing state.

We better related the Maki-Thompson model with the gossip protocol, which provides a stronger argument in justifying the original transitions. Particularly, in Sec. II A we wrote the following explanation

However, the relevance of rumor models is not limited to social context. Indeed, this class of models inspired and constituted the theoretical basis for the so-called gossip protocol [6], which is a powerful paradigm used in the design of decentralized distributed protocols that are reliable and efficient [7]. The applications of this protocol are quite wide, including Peer-to-Peer Networks (P2P) [7, 11] and specific tasks as failure detection [7, 10], to implement garbage collection [7, 11], to compute aggregate information [7, 12], and to allocate resources [7, 13]. Moreover, specific real-life examples, include the Gnutella P2P Network [9] and cryptocurrencies networks as Bitcoin [14, 15] or Ethereum [16] and their derivations. Although these examples are easily relatable to our daily lives, surprisingly, this class of processes is much less explored than other stochastic dynamics [17, 18].

Comment 1.3

Incidentally, the term “ignorants” to describe those as yet unexposed to the rumor seems odd, almost offensive. Why not use “susceptibles” or “unexposed” to label this segment of the population?

Reply 1.3.— We understand the point raised by the reviewer, and we agree that “susceptibles” or “unexposed” are better terms to name them. However, this nomenclature is commonly used in papers in which these models have been investigated over the years. For example, see the references [5, 6, 7, 8, and 10] of our manuscript. We are concerned about changing the term used, which could cause misunderstanding in the scientific community. Although this word may sound inelegant, we are using it here in the sense of awareness about a particular thing. We appreciate the reviewer’s care, and we include in the text a better explanation of the terminology used.

Action taken 1.3.— In the first paragraph of the introduction section, we included the sentence:

To be more specific, the terminology ignorant is widely used in rumor models to indicate someone unaware of the news in the sense that she/he did not hear about the news, this means someone who has not yet been exposed to the rumor.

Comment 1.4

Summing up, I believe the authors should provide a stronger motivation for the MT models and their relation to models of epidemics, to dynamic percolation [see P Grassberger, *MATHEMATICAL BIOSCIENCES* 63: 151-172 (1983)], fashions, opinions, and chain letters.

Reply 1.4.— We thank the reviewer for these suggestions. We improved the motivation for our work in two ways. First, we improved the text to better relate rumor models and epidemic models. Second, we better justified the annihilation mechanism by explicitly mentioning the “reluctance to spread stale news” from the original Daley-Kendall model.

Regarding the relation to dynamic percolation, we found it very interesting but also not trivial. Notice that in [1], the author explicitly mentions that the general epidemic process corresponds to neither the site nor bond percolation. However, under the approximation that time is discrete and equally spaced as $T = \frac{1}{\delta}$ (the average recovery time), the process can be mapped into a bond percolation problem. On the other hand, by assuming that every spreader hits all susceptible neighbors simultaneously, then we have a site percolation problem. In both cases, the percolation probability can be extracted and depend only on the spreading and recovery rates. However, in the rumor model, the percolation probabilities are dependent on the number of neighbors that are aware of the information, thus making the mapping much more difficult.

We believe that exploring the relationship between rumor models and dynamic percolation might be relevant for a better understanding of the universal properties of rumor spreading. However, we believe that this is out of the scope of the paper and might even make the paper less clear. Thus, we included a sentence at the conclusion, pointing out this relationship but referring to it as future work. We did not feel confident about making a simpler comment as we could introduce inaccurate information.

[1] P Grassberger, *MATHEMATICAL BIOSCIENCES* 63: 151-172 (1983)

Action taken 1.4.— In the conclusion we added the following sentence

We identified two main future research lines that might give an additional understanding of rumor dynamics. First, we believe an in-depth analysis of the universality class of rumor models, as well as understanding its relationship with dynamical percolation [55], might provide some additional mechanistic explanations for the observed phase transition. Second, we believe that the Geometric Singular Perturbation Theory, as applied for the SIRS [56], might provide additional analytical insights, especially in the lower λ regime.

Comment 1.5

Absorbing States

Once the number of spreaders is zero, the rumor is fated to die. In the original MT model, as in an SIR epidemic, this end state is nonunique. Here, the authors introduce a (spontaneous) transition from stiffer to ignorant (analogous to changing from SIR to SIRS) to eliminate the multiplicity of the absorbing state. But this would appear to change the universality class, just as SIR (with no active steady state) belongs to dynamic percolation whereas SIRS or SIS (which do admit active steady states) belong to directed percolation. The original MT model is intrinsically a spreading process, whereas the modified MT model can maintain activity indefinitely. The authors' motivation for this alteration (that a unique absorbing configuration allows them to use quasistationary simulations) seems neither correct nor desirable, if it changes the universality class.

Reply 1.5.— We are very grateful to the reviewer for pointing out this important issue. In reference [1], Sander and collaborators showed an agreement among the critical exponents of different models based on the contact

process model and running on top of different kinds of networks. In short, they concluded that the universality class of the contact process by a heterogeneous mean-field approach could be conjectured to include other models similar to the contact process, independent of the number of absorbing states. In fact, the universality class seems to depend only on the locality of the interactions.

Bancal and Pastor-Satorras [2] had also shown, using the forest-fire model, that models belonging to the SIS universality class for heterogeneous networks can present a finite or infinite number of absorbing states, depending only on the way the activity spreads over nearest neighbors. In fact, this seems feasible since when we compare the SIS and CP models, we observe that they belong to the directed percolation class, according to the Janssen-Grassberger conjecture, as mentioned by the reviewer. This happens because both models are equivalent in homogeneous networks. In the CP model, a node divided its infection equally among all neighbors, while in the SIS model, a node infects all its neighbors with the same strength. However, in complex networks, heterogeneity affects both models, and, at the heterogeneous mean-field approach, they have different critical exponents, and, consequently, the SIS model does not belong to the directed percolation class anymore.

In essence, we hope that the reviewer shares with us our previous argument regarding the motivation for introducing a spontaneous transition from stiffer to ignorant in our model since this adjustment does not seem to affect its universality class. We thank the reviewer again for emphasizing this crucial point that certainly deserves to be included in the manuscript.

[1] Sander, R. S. and Ferreira, S. C. and Pastor-Satorras, R., Phase transitions with infinitely many absorbing states in complex networks, *Phys. Rev. E* 87 (2013).

[2] Bancal, JD., Pastor-Satorras, R. Steady-state dynamics of the forest fire model on complex networks. *Eur. Phys. J. B* 76, 109–121 (2010).

Action taken 1.5.—In the last paragraph of the introduction, we enlighten this subject by adding the sentence:

It is important to emphasize that, although the introduction of a spontaneous transition from stiffer to ignorant affects the number of absorbing states in the model, this should not change its universality class. As discussed in references [40, 41] for the contact process and the SIS model, respectively, the authors showed that the universality class seems to depend only on the locality of the interactions and not on whether there is a single or infinitely many absorbing states.

Comment 1.6

Network structure

The term "random regular network" does not appear to be standard in the literature, though a recent review article by one of the authors gives to understand that this is an Erdos-Renyi graph. In any case, a more precise definition and some details of the construction algorithm as needed. An illustration of a typical network, and of the progress of rumor spreading on it, would be highly desirable. (Videos of illustrative examples would also be helpful!)

Reply 1.6.— We agree with the reviewer that the definition and some details about the random regular network should be included. In addition, a description of uncorrelated power-law networks is also required. In fact, in a random regular network, all vertices have the same connectivity $\langle k \rangle$ while the connections are made at random, avoiding both self and multiple connections. This explanation has been included in the text.

We thank the reviewer for the great idea about the video to illustrate the spread of the rumor on the network. Videos are now attached as supplementary material to the manuscript.

Action taken 1.6.—An explanation about random graph has been added in Sec. II B as

In this network, all nodes have the same number of neighbors $\langle k \rangle$, and the connections between them are made at random, avoiding both self and multiple connections.

Moreover, the explanation about uncorrelated networks has been added in Sec. II C as (please, see also footnote 3)

To understand its effects in the phase transition, we analyzed the critical properties of uncorrelated random networks with a power-law degree distribution, $P(k) \sim k^{-\gamma}$, generated by the algorithm proposed by Catanzaro et al. [46], named uncorrelated configuration model. These networks are important from a numerical point of view because it is possible to test the behavior of dynamical systems whose theoretical solution is obtained only in the absence of correlations.

Finally, we made three videos exemplifying the subcritical, critical and supercritical regimes. The videos were made in a lattice for visualization purposes. We hope to attach them as supplementary materials.

Comment 1.7

Citations

The topic of rumor spreading on networks has been discussed in many recent studies. The authors need to enlarge and diversify their citations to the recent literature.

Reply 1.7.— We completely agree with the reviewer. Although there are not so many recent works about the traditional DK and MT models, there are many interesting and current works discussing the subject and proposing new compartmental models with very relevant applications.

In addition to the references focusing on the theoretical analysis of the DK and MT models, we diversified our citations by including a discussion about the gossip protocol (see also Comment 1.2). Note that, with these references, we are diversifying our citations, including computer science references, which we hope will widen our paper's audience.

Action taken 1.7.— We added new and diversified references in the introduction. Specifically, we added the following sentence

Recently, there have been several works about spreading of rumors investigating new compartmental models with practical applications such as, for example, fake news or other dissemination processes in online social networks [33-38]. However, these works are concerned with proposing more realistic models without focusing on the phase transition analysis of classical models such as the MT model.

In addition to the above sentence, other references were also added throughout the text. For instance, by including the introductory paragraph about applications and the gossip protocol, we incorporated references in the field of computer science, which were missing in the previous version.

Comment 1.8

In summary, the authors have demonstrated that the MT model, in its original version, exhibits a phase transition. This result, which corrects a longstanding misconception based on a simple mean-field analysis, seems the most significant finding in the paper, and certainly merits publication. I would support acceptance of a manuscript revised to address the issues cited above.

Note: I'm attaching an annotated version of the paper.

Reply 1.8.— We are pleased to know that the reviewer thinks our paper “certainly merits publication”. We thank the reviewer again for all their comments that have significantly improved our work. We also appreciate their kindness in attaching an annotated version of the paper. However, we wonder whether this was the correct file, as we found only one correction.

DETAILED REPLY TO THE REPORT OF REVIEWER 3

Comment 3.1

The authors present interesting observations about the rumor spreading model of Maki Thompson and their “delta” variation. The introduction is very nice; the Fig. 1 & 2 are crucial and new. Their paper is publishable, in my opinion, after a serious rewriting. Below, I motivate this recommendation. Even though the mathematical description of the model (Sec II.a) is poor, I believe that their model is correct: At each time t , each node is either Ignorant (susceptible) X , Spreader (infected) Y or Stifler (recovered) Z . Let me explain my confusion. I believe that X , Y , Z are not Bernoulli random variable, but they describe the state of node i . Indeed, if X , Y , Z are Bernoulli random variables, what is the physical meaning of $X = 0$, $Y = 0$ and $Z = 0$ (only $X = 1$, $Y = 1$ and $Z = 1$) is explained. Later on, $Y = 0$ is mentioned and infinitely many absorbing states (which is really confusing because a Markov process on a finite graph has rarely more than 2 absorbing states). Their notation is slightly confusion, because the strength of continuous-time Markov processes lies in the fact that at any time, only one transition occurs. Here, I believe that, within each time step Δt , and with the authors’ definition of X_i , Y_i and Z_i , multiple values change for each transition. Nevertheless, I think that a precise description can be resolved. The corresponding Markov chain still consists of 3^N states (like in SIR, even though this is never mentioned explicitly). The model is now explained in the thermodynamic limit (i.e. for infinitely large graphs), while reality and their simulations are finite. I think that it is better to explain the model for finite size and occasionally talk about the limit.

Reply 3.1.— We thank the reviewer for their thorough evaluation of our paper. We admit that our notation was a bit confusing, and we tried to improve the description of our model and the presentation of our findings. We implemented all the reviewer’s suggestions, clarifying the points that generated the above-mentioned confusion. We believe that our paper has improved, both in quality and presentation. Thus, we hope that this new version is suitable for publication.

First, regarding the random variables X, Y, Z , we clarify that they are associated to each node and, for each individual i they must respect $Y_i + X_i + Z_i = 1$. In this case, only one variable is allowed to be one. As they can only be 0 or 1, they fit into the definition of a Bernoulli random variable. Concerning their meaning, for instance, $Y_i = 1$ implies that the node i is a spreader. Complementary, the meaning of $Y_i = 0$ is that node i is not a spreader. The same also applies to the other variables. Thus, due to the relationship $Y_i + X_i + Z_i = 1$, we unequivocally define the states of our nodes by the vector (X_i, Y_i, Z_i) . We remark that this formulation is different than defining the states as $\eta_i = \{0, 1, 2\}$, where, for example, $\eta_i = 0$, $\eta_i = 1$, and $\eta_i = 2$ would mean that node i is ignorant, a spreader or an stifler. Both approaches are equivalent. However, using the Bernoulli random variable definition is more convenient as the expectation of a Bernoulli random variable is its probability.

Second, we also clarify that both our analysis and simulations were done following the continuous-time Markov chain formalism. Indeed, as pointed by the reviewer, only one state changes at a given event. Additionally, the inter-event times are also a random variable that depends on the state of the nodes.

Regarding the number of absorbing states, we clarify that we are referring to the number of micro-states. Notice that each micro configuration in which we have no spreaders is an absorbing state for the standard MT model. However, in the modified model, the only possible absorbing state is $(1, 0, 0)_i$ for all $i = 1, 2, \dots, N$. To conclude, notice that $Y_i = 0$ for all $i = 1, 2, \dots, N$ is a necessary condition for a state to be an absorbing state in both models. In other words, in both models, in the absorbing state, there are no spreaders. However, in the MT model, the individuals might be stiflers or ignorants, while in the modified version, we just have ignorants.

Finally, as each individual has three possible states, our process has 3^N possible micro-states. As a consequence, it can be exactly solved by a set of 3^N equations.

Action taken 3.1.— To avoid the confusion regarding our notation, we rewrote the local rules definition as

Denoting the state of the node i as $(X_i, Y_i, Z_i)_i$, the above described local rules are expressed as

$$\begin{aligned} (0, 1, 0)_i + (1, 0, 0)_j &\xrightarrow{\lambda} (0, 1, 0)_i + (0, 1, 0)_j, \\ (0, 1, 0)_i + (0, 1, 0)_j &\xrightarrow{\alpha} (0, 1, 0)_i + (0, 0, 1)_j, \\ (0, 1, 0)_i + (0, 0, 1)_j &\xrightarrow{\alpha} (0, 0, 1)_i + (0, 0, 1)_j, \\ (0, 0, 1)_i &\xrightarrow{\delta} (1, 0, 0)_i. \end{aligned}$$

A graphical representation of these transitions is presented in Fig. 1.

We added a discussion about the complexity of three-state dynamics and justified our choice of first discussing the thermodynamic limit and then focusing on finite systems in the last paragraph of Sec II A:

We remark that both the MT model and our modified version are represented by a continuous-time Markov chain with 3^N possible micro-states. Thus, it is theoretically possible to write the infinitesimal generator, leading to a linear system that solves these processes exactly. However, this is not possible in practice due to the prohibitive computational cost. Notice that this is the case of the SIR model or any three-state dynamics. Note also that the concept of phase transitions is only valid in the thermodynamic limit. However, as it is commonly done in complex systems [17, 18, 21, 24, 26], we also use the term phase transition in finite systems as we observe two different scaling behaviors, as mentioned above. Despite the fact that all our simulations are performed in a finite system, for this theoretical reason, most of our analysis implicitly assumes the thermodynamic limit. The exception here is Sec. II C 3, which follows an individual-based approach.

[7] G. F. de Arruda, F. A. Rodrigues, and Y. Moreno, *Physics Reports* 756, 1 (2018)

Comment 3.2

Suggestions:

a) the MT model is relatively unknown, while the epidemic theory is huge. Hence, I would draw the analogy between SIS, SIR and SIRS much better and clearly point (a) what is the same and (b) where the difference lies. Next to the “chemical” presentation in II. A, a figure of the transitions in node i may be useful. b) Move the asymptotic analysis to the appendix or supplementary material. Why? First, eq. (10) are SIRS equations in first-order mean field. Only the last equation deviates from the many analyses of SIR (e.g. see the book of Kiss, Miller & Simon). The notation x, y, z used is due to Kermack-McKendrick. Second, the only point is to show that, as mentioned frequently in the paper, that a first-order mean-field analysis does not explain Fig. 1 & 2.

Reply 3.2.— We thank the reviewer for their comments and suggestions. We improved the discussion about the relationship between rumor and epidemic models. We included a short historical fact, acknowledging Goffman and Newill for pointing that out. Most importantly, we emphasized the similarities and differences between both models.

Regarding the suggestion about moving the asymptotic analysis to the appendix, we moved the derivations to the Methods section but kept a summary description of these results in the main text. Although the details of this result might be interesting for just a relatively restricted number of researchers, the conclusions are still important and appealing to a wider public. We believe that the results of the asymptotic analysis show a fundamental limitation of the mean-field approach. Thus, by separating our analysis into two sections (main text + methods), we hope to appeal to both researchers that are interested in the theoretical aspects and also more applied ones.

In the new version, we left equations (10) and (11) as part of the main paper. We would like to point out that these equations are sufficiently different from the SIR or SIRS. Note that, in our rumor model, the first-order

mean-field equations depend on the products $y_i y_k$ and $y_i z_k$, which are not present in the epidemic cases. For the sake of an example, it is possible to obtain the critical point intimations for the SIR or SIRS just by neglecting the product of the probabilities of infected individuals, thus arriving at an eigenvalue problem. Here, such an approach is not possible.

Action taken 3.2.— We better explained the relationship between epidemic and rumor models in the introduction as

Despite having different purposes, rumor and disease spreading models are mathematically similar. Goffman and Newill were the first to notice the analogy between the spreading of a disease and information dissemination [19, 22]. Both processes are typically modeled by compartmental models, where the population is divided into mutually exclusive and exhaustive classes. Also, the spreading of disease and rumor processes are often modeled in the same way (this is the case of SIS, SIR, SIRS, MT, and DK models). However, the removal mechanism is usually different. In rumor models (DK and MT), this mechanism is driven by the contact between individuals aware of the information (spreaders and stiflers), while in the epidemic case, it is spontaneous. This mechanism was initially proposed in [19] and was motivated by the hypothesis that an active spreader stops telling the rumor because she/he learns that it has lost its ‘news value.’ The authors called this mechanism the “reluctance to tell stale news.” Conversely, if one considers that the removal would be only due to a ‘forgetting’ mechanism (spontaneous), this process would follow the same equations as a SIR epidemic spreading model.

We included a figure with the transitions for both the Maki Thompson and our modified model.

We rewrote the asymptotic section in the main text, focusing on the results and only providing a description of the main ideas about the performed analysis. We also moved the asymptotic derivation to the Methods section, which is at the end of the paper in the Journal’s format.

Comment 3.4

Specific small remarks to improve the paper:

- In the abstract, it says: ...”trusted or not”. I think that the authors mean ”...trustworthy or not”.

Reply 3.4.— We agree with the reviewer that “trustworthy” is better suited in this context.

Action taken 3.4.— We changed accordingly.

Comment 3.5

- In the introduction, the authors states that ”...we were able to show that the mean-field approaches cannot predict a phase transition.” The statement is only correct while considering first-order mean-field methods. Perhaps higher order mean-field approximation may capture this property.

Reply 3.5.— The reviewer is absolutely right. We agree that it is best to make it clearer.

Action taken 3.5.— We included the term ”first-order” in the sentence.

Comment 3.6

- I do not understand the second part of this statement: “Notice that δ can be interpreted as either as a forgetting

mechanism or, for evolving rumors, individuals who do not know the rumor's current state.”

Reply 3.6.— We admit that this statement was a little unclear. We meant that in addition to being a forgetting mechanism, the δ parameter could indicate an evolving rumor, in the sense that someone who is spreading a rumor A and finds out about an update of this rumor or even a new one B, he/she loses interest in propagating rumor A (forgetting mechanism) and starts propagating rumor B, and so on. Admittedly, this assumption implies a series of approximations regarding the rumor. However, the essence of the process is captured by the forgetting mechanism (with parameter δ).

Action taken 3.6.— We better explained this statement in the text.

Comment 3.7

Sec. II. B: “the fraction of stiflers ... independent of system size”? Strong statement and incorrect, I think. There is dependence, but, perhaps, weak. Next sentence: “these results indicate a phase transition” (why?) “peak of lifespan diverges...limit”: add reference, as well as, in the sentence “this behavior is compatible with a second-order phase transition”.

Reply 3.7.— We thank the reviewer for the meticulous revision and comments. We do agree with the comments. The reason we used the word independent is that, as we increase the system size, the curves should converge to the curve obtained in the thermodynamics limit. However, a more precise exposition is explicitly saying weak dependency. We clarified these points and better explained the phase transition.

Regarding the references, we also agree that they are missing, and we included them in the revised version.

Action taken 3.7.— We changed the word “independent” by the sentence weakly dependent on the system size (see Fig. 1 for an example).

We substituted the sentence “These results indicate a phase transition in the MT model as a function of λ ” by the following explanation.

Note that, as we increase the system size, for a small λ regime, i.e., $\lambda < \lambda_c$, where λ_c is the critical point, the fraction of stiflers goes to zero. On the other hand, for the larger λ regime, i.e., $\lambda > \lambda_c$, the curves converge to the value of the fraction in the thermodynamic limit. This behavior suggests a phase transition in the MT model as a function of λ .

Comment 3.8

The authors devote a lot of time/space to the subcritical regime, where lambda is very small. In my opinion, the observed power-law behaviour is due to the process with rate alpha: Spreader nodes can only become stiflers while being in contact with a spreader or with another stifler. If the network is initiated with a single spreader node (which is the case here), then that node is only connected to ignorant nodes. Considering the limit of small “infection” rates lambda, the spreader node must infect at least one neighbouring ignorant node before the “curing” process can take place. Thus, we need to wait for one lambda-event to take place (which is slow, because lambda is small) whereafter two alpha events take place (which are fast compared to lambda) and then the process has converged to the steady state. I think the small lambda regime is currently not well explained in the paper. Finally, I think that an analysis for small lambda (see e.g. Geometric Singular Perturbation Theory, <https://arxiv.org/abs/2011.02169>) is possible, although it is not necessary here at all.

Reply 3.8.— We thank the reviewer for the comment. We do agree that the reader can greatly profit from a better

explanation about the small λ regime. In the revised version, we provided a mechanistic explanation, discussing the role of each event and their respective expected time on T (equations (8) and (9) in the revised paper).

Regarding the Geometric Singular Perturbation Theory and the cited reference, we do believe that this approach could help our analysis. However, further analysis with a careful development is necessary, where the similarities and differences between the SIRS and our model should be meticulously dealt with. Thus, we believe it is out of the scope of the paper. Indeed, we believe that such an analysis could originate an independent paper. As such development is an interesting research line, we wrote a sentence about this possible future work that we hope to explore or to be explored by other researchers.

Action taken 3.8.— To better explain and discuss this phenomenon, we added the following paragraph in Sec. II C 2:

It is noteworthy that, in the subcritical regime, the time to reach the absorbing state is dominated by the time the system has to wait before a spreading event happens. In order to reach the absorbing state, whenever we have a single spreader, it first needs to inform a neighbor and, only then, the process will be allowed to reach the absorbing state. Notice that, after the spreading event, the stifling events are much faster than the spreading ones. As the spreading events are the slowest ones in the subcritical regime, the rate of these processes dominates Eq. (8), as shown in Eq. (9). Also, frustrated trials to reach the absorbing state are possible. However, in this case, we are only repeating the above-described process.

Moreover, we added a comment about the Geometric Singular Perturbation Theory in the conclusion as:

We identified two main future research lines that might give an additional understanding of rumor dynamics. First, we believe an in-depth analysis of the universality class of rumor models, as well as understanding its relationship with dynamical percolation [55], might provide some additional mechanistic explanations for the observed phase transition. Second, we believe that the Geometric Singular Perturbation Theory, as applied for the SIRS [56], might provide additional analytical insights, especially in the lower λ regime.

Comment 3.9

In Sec II.B.3 the authors explain their asymptotic analysis but the statement "the proof of existence of a critical point is impossible in a first-order approximation" is too strong, even though I agree a proof will be hard.

Reply 3.9.— We thank and agree with the reviewer for this comment. In our asymptotic analysis, for some regimes, we assumed a random regular structure in order to obtain an analytical solution. So, in the case of random regular networks, we are not able to find a transition using the first-order mean-field approach. However, we are unable to guarantee that the transition can not be found in the first-order mean-field approach for a general structure. To be clearer and more precise, we changed this sentence.

Action taken 3.9.— We substituted the word "impossible" with the expression "very hard or even impossible."

Comment 3.10

Sec. II. C.1: eq. (3) seems to hold in a k -array tree. A bit more explanation would be helpful. It seems that standard branching process theory is applied?

Reply 3.10.— We thank the reviewer for this comment.

Yes, it holds for k -ary trees. In fact, the MT model is exactly described by a branching process on an infinite tree. In the text we clarified this point and included the book [1] as a reference. To be precise, Eq. (4) is a consequence of the extinction probability in a branching process. This concept is formally discussed in Theorem 1 in Chapter I of [1], which was duly cited in the main text.

[1] K. Athreya and P. Ney, *Branching Processes, Die Grundlehren der mathematischen Wissenschaften* (Springer Berlin Heidelberg, 1972), ISBN 978-3-540-05790-1.

Action taken 3.10.— We clarified that we are indeed using branching theory in the following sentence, before Eq. (3):

In the case of the MT model on an infinite tree, the rumor propagation is exactly described by a branching process [45]. By averaging over the degree distribution, the condition that establishes the transition between a phase where the rumor dies out with probability 1 and a phase where there is a non-zero probability of infinite propagation [45, Theorem 1, Chap. I] is ...

After the presentation of equations (3) and (4) we added the following explanation:

In other words, Eq. (3) estimates the expected number of newly informed individuals as a result from a single initial spreader event by weighting the probabilities that the rumor stops (accounted by item (a)) or that it continues the spreading (accounted by $q_k(i+1)$). Moreover, the transition can be obtained from the condition at which the expected number of newly informed nodes is larger than one, i.e., are more likely to spread the information than to stop the spreading. Note also that this approximation is based on the fact that only a single event occurs at a time, which is a property of continuous Markov chains.

Comment 3.11

- Explain or give a reference to “first-step approximation” (see e.g. the books of Karlin & Taylor)

Reply 3.11.— Thank you for pointing this out. It was a typo. We were referring to a first-order approximation, where we evaluate the competition of the spreading against the stifling mechanisms.

We also thank the reviewer for this point, as it helped us to notice that our approximation was slightly off. We also clarify that the approximation in the first version neglected the two stifling processes and considered just one stifling process. In this new version, we corrected the approximation. However, this does not change any of our results or conclusions.

Action taken 3.11.— We changed the term to one-step process to first-order approximation and better explained the approximation in the sentence

In the same figure, we also present the naive estimation, considering a first-order approximation, given as $\lambda^* = \frac{\alpha}{\langle k \rangle - 1}$. Note that this expression accounts for the competition between the spreading processes, with the rate $2(\langle k \rangle - 1)$, and the annihilation, with the rate 2α .

Comment 3.12

- What are frustrated trials?

Reply 3.12.— A frustrated trial is when one tries to fall into the absorbing state and is not able to.

Action taken 3.12.— We added the following definition at the first time we used this concept:

Regime	HMF		QMF		Monte Carlo	
	$2 < \gamma < 3$	$\gamma > 3$	$2 < \gamma < 3$	$\gamma > 3$	$2 < \gamma < 3$	$\gamma > 3$
SIS	Vanishing	Non-null	Vanishing	Vanishing	Vanishing	Vanishing
SIR	Vanishing	Non-null	Vanishing	Vanishing	Vanishing	Non-null
SIRS	Vanishing	Non-null	Vanishing	Vanishing	Vanishing	Non-null

TABLE I. Critical point behavior for epidemic processes in power-law networks.

Here, a frustrated trial is defined as the scenario in which the system needs one event to be forced into the absorbing state, but a spreading event takes place instead, thus avoiding the absorbing state.

Comment 3.13

- “At odds with power-law networks ... vanishing critical point for any ... gamma”: I think that each network with an average degree that increases unboundedly with size N has a “SIS/SIR vanishing epidemic threshold” in the limit $N \rightarrow \infty$.

Reply 3.13.— Taking the example of networks with power-law degree distribution, $P(k) \sim k^{-\gamma}$, the average degree (and all the higher moments) will vanish if $\gamma \leq 2$. For $2 < \gamma < 3$ the mean exists but the second moment diverges.

A SIS process has a vanishing critical point in the thermodynamic limit, regardless of γ , as proved in [1]. However, the same is not true for the SIR model. Using the message-passing approach [2], it is possible to exactly show that the critical point for tree-like structures will depend on the inverse of the leading eigenvalue of the non-backtracking matrix. Also, this is expected to be finite for $\gamma > 3$ [3] (more precisely, see Eqs. 39 and 19 in [3]). In addition to these theoretical predictions, in Ref. [3], they showed an example of a vanishing behavior for $\gamma = 2.1$ (Fig. 5 (a) of Ref. [3]), while they noticed a finite critical point for $\gamma = 3.5$ (Fig. 5 (b) of Ref. [3]). Both cases were correctly predicted by the message-passing approach.

Next, regarding the SIRS model, while HMF and QMF theories predict the same behavior as the SIS model, the main result of [4] is that the effect of even a small amount of waning immunity is able to restore a finite threshold in power-law networks with a degree distribution with exponent $\gamma > 3$.

The above results are summarized in Table I.

[1] S. Chatterjee and R. Durrett, *Ann. Probab.* **37**, 2332 (2009).

[2] B. Karrer and M. E. J. Newman, *Phys. Rev. E* **82**, 016101 (2010).

[3] W. Wang, M. Tang, H. E. Stanley, and L. A. Braunstein, *Reports on Progress in Physics* **80**, 036603 (2017).

[4] S. C. Ferreira, R. S. Sander, and R. Pastor-Satorras, *Phys. Rev. E* **93**, 032314 (2016).

Comment 3.14

Sec. III: “Crucially, this expression ... local correlations ...: I assume that “this expression” is eq. 3? Since eq. 3 is a recursion for the mean of a random variable in a branching process, I do not understand where “correlations” enter? A mean of a random variable does not incorporate correlations (i.e. joint probabilities). Also, “if $\delta > 0$, the rumor might still wander around ... before die-out” and the next sentence can be written more clearly to convey the message.

Reply 3.14.— We thank the reviewer for their comments. We agree that the term correlation was misplaced. Here we were referring to the local feedback effects. The key feature we capture in the tree-based (branching process)

approximation is that spreading events introduce a feedback loop as they increase the probability that the initial spreading node is stifled. In the mean-field calculation this effect is averaged out which leads to underestimating the local rate of stifling. We changed this sentence in the conclusion.

We agree that our explanation about the fact that the rumor can wander around before disappearing was insufficient. We wrote an explanation about this effect at the end of Sec. II C 2. The main idea behind this effect is that, in this regime, we always have a very small number of individuals that are aware of the rumor but, at the same time, these individuals might change. In summary, as we need at least two individuals that are aware of the information (spreaders or stiflers), we might have that the stiffer forgets before converting the other spreader into a stifer. In this case, the spreader has to inform someone. However, the spreader is not necessarily the same as before. Thus this process makes the information travel.

Action taken 3.14.— We changed the above mentioned sentence to

Crucially, the tree-based (branching process) approximation, Eq. (3), explicitly accounts for local feedback effects that are ignored by the first-order mean-field approximations that fail to capture the transition. The key feature we capture in this approximation is that spreading events introduce a feedback loop, as they increase the probability that the initial spreading node is stifled. In the mean-field calculation, this effect is averaged out, which leads to underestimating the local rate of stifling.

Regarding the comment about the “the rumor might still wander around,” we included a paragraph in Sec. II C 2 that reads as

Note that in our modified model, $\delta > 0$, the rumor might still wander around in the network before die-out. For every frustrated attempt to reach the absorbing state, the spreader individual might be a different one. In this way, although the rumor is fated to disappear, during the transitory, many different individuals might be informed about the rumor before it dies out. This effect is a consequence of the “reluctance to tell stale news” (the annihilation by contact) with the forgetting mechanism. The first mechanism imposes that at least two spreaders are necessary to reach the absorbing state, while the second mechanism enforces a single absorbing state.

Comment 3.15

Minor issues:

- Spelling: E.g. “In this case, as the system size increases, the number of absorbing states also increaseS.”

Reply 3.15.— Thanks for the note. We corrected the text.

Comment 3.16

- General: try to avoid “this and that”, but be precise in what “this” mean. Each “this” implies that a reader must “substitute this” by the relevant quantity.

Reply 3.16.— We thank the reviewer for raising up this point. We reviewed the text carefully and we tried to be more specific.

Action taken 3.16.— Revising the text, some “this” was replaced to make the text more precise.

Comment 3.17

- Fig 1: legend for black curve show 1×10 , but it should probably be 1×10^3 (see fig. 2)

Reply 3.17.— Yes. Thank you for pointing out this typo. We fixed it.

Comment 3.18

- Which method is used by the authors to generate random regular graphs? Also, we remark that the class of random regular graph is a rather small, special class. There are less random regular graphs than co-spectral graphs!

Reply 3.18.— We apologize because, in fact, the method for generating random regular networks was not really clear in the text. The definition and some details about the random regular network should be included. In fact, in a random regular network, all vertices have the same connectivity $\langle k \rangle$ while the connections are made at random, avoiding both self and multiple connections. This explanation has been included in the text.

Action taken 3.18.— We added the following explanation about random graph

In this network, all nodes have the same number of neighbors $\langle k \rangle$, and the connections between them are made at random, avoiding both self and multiple connections.

Comment 3.19

- Fig. 3: λ_c is not defined.

Reply 3.19.— We fixed it and set it in the caption. We thank the reviewer for the observation.

Comment 3.20

- Fig. 4: what is an “uncorrelated” power-law network?

Reply 3.20.— An uncorrelated network is a network whose nodes are connected to each other regardless of their degree, that is, a node with degree k can connect to any other node with degree k' without any preference.

Action taken 3.20.— We agree with the reviewer that this term was not explained in the text. So, we added a brief explanation with the appropriate reference at the first column of page 5, as well as a footnote.

Comment 3.21

- Fig 6: the horizontal time axis better uses $10^1, 10^2, 10^3$ as time units - Fig 6: the caption says $\lambda \gg \delta$, but I think it should be the reverse: $\lambda \ll \delta$.

Reply 3.21.— We thank the reviewer for these suggestions. We do agree that $10^1, 10^2, 10^3$ as time units make the figure clearer. Moreover, we thank the reviewer for pointing out the typo.

Action taken 3.21.— We changed the figure and corrected the caption.

REVIEWERS' COMMENTS

Reviewer #4 (Remarks to the Author):

Most of the concerns of the previous referees have been adequately addressed. The manuscript contains sufficiently novel and significant results to warrant publication.

I only have some concerns regarding Section II.C.3 (page 8), which I find quite obscure. I am not very familiar with asymptotic analysis, but it is likely that this applies also to part of the readership.

- Three possible regimes may apply. What determines which of the three actually holds?
 - It would help to show explicitly that the same analysis applied to the SIRS model predicts the existence of a transition in that case.
 - λ_{\max} is not defined in the main text (although it is defined in the Methods).
- Isn't the largest eigenvalue of a k -regular graph equal to k ?

In addition, I have many suggestions for amendments to correct details in the manuscript that considerably reduce its readability.

- Starting from the abstract, the transition found in the MT model is denoted as second-order. A few lines before the mean-field approach used in previous papers is denoted as first-order. This is confusing, as the two "orders" are not related. I suggest to denote the transition as "continuous".
- Brackets are used to denote averaging over realizations of the dynamical process and also averaging over the degree distribution. This is a bit confusing.
- The statement "causing impact on more than 50 years of investigation" is definitely excessive.
- Page 2, col. 2: weather  whether
- Page 3, beginning of col. 1: This limitation can be related to the non-exponential decay present in the sub-critical behavior" "Non-exponential decay" suggests a temporal decay. If I understand correctly here a non-exponential decrease with λ is intended. Is the non-exponential nature so important? Isn't the decrease (instead of the increase) the crucial point?
- Page 4, col. 1:
"To characterize the MT model, we can simulate the process beginning with many different initial conditions"
What are the initial conditions used in Figs. 3, 6 and 8?

- Fig. 3: Is the decay for small λ close to $1/\lambda$ also in this case? Can you put a straight line with slope -1 in the figure?
- Fig. 4: The meaning of λ^* should be specified in the caption. What does $q(1)$ mean in the legend? Doesn't it mean that the value plotted is λ_c for which $q(1)=1$ in Eq. (4)? If this is the case, the indication " $q(1)$ " in the legend is misleading.
- Pag. 5, col. 1:
where the lifespan increases as λ increases 
where the lifespan decreases as λ increases
- Can you specify in detail the form of Eq. (4) for the random regular case ?
- Fig. 5: what is the value of δ ?
- Pag. 6, col. 1: how is λ^* derived? Does it come from Eq. (4) under some approximation?
- Fig. 6: How has the orange line been derived? Fitting to what? The number of digits of the exponent is excessive.
- Eq. (6): $k \rightarrow \langle k \rangle$
- Pag. 7, col. 1:
"the time to reach the absorbing state, $T_{\{abs\}}$.
This quantity was called τ in Eq. (2).
- Pag. 7: $A +$ is missing in Eq. (9).
- Pag. 10, col. 1, after Eq. (12):
 ρ_Y is the quasi-stationary distribution 
 ρ_Y is the quasi-stationary density
- Supplementary Information is not mentioned in the manuscript.

From subcritical behavior to a correlation-induced transition in rumor models
(NCOMMS-21-29679A)
Reply to the Reviewers
(Dated: April 28, 2022)

DETAILED REPLY TO THE REPORT OF REVIEWER 4

COMMENT 2.1

Most of the concerns of the previous referees have been adequately addressed. The manuscript contains sufficiently novel and significant results to warrant publication.

Reply 2.1.— We thank the reviewer for their thorough evaluation of our work and for their constructive comments.

COMMENT 2.2

I only have some concerns regarding Section II.C.3 (page 8), which I find quite obscure. I am not very familiar with asymptotic analysis, but it is likely that this applies also to part of the readership.

- Three possible regimes may apply. What determines which of the three actually holds?

Reply 2.2.— In the asymptotic analysis we are interested in the limiting behavior. In this case, we assume that we can expand a variable in a series and neglect higher-order terms (e.g., $y_i \sim y_i^{(1)} \epsilon^c + O(\epsilon^{2c})$, where ϵ is small and $y_i^{(1)} \in O(1)$). Note that, here we are interested in the lowest powers of ϵ as the higher powers do not change the limiting behavior. So, the regimes are defined as a relationship between the powers of different quantities (the leading terms). Observe that, in order for the mean-field set of equations to be consistent we have to be in one of the three identified regimes: (i) $0 < c < k$, where $c = m - n$ and $k = 2m - n$, (ii) $c = k$, where $m = 0$ and $n = -k$, and (iii) $0 < k < c$, where $c = -n$ and $k = m - n$. The first regime implies that the probability that node i is a spreader follows $y_i \sim \frac{\lambda}{\alpha}$. The second regime implies that $y_i \sim \frac{\lambda}{\alpha(\lambda\Lambda_{\max} + 1)}$, while the third regime implies that $y_i \sim \frac{1}{\alpha\Lambda_{\max}}$. We remark that we can go from one regime to another by controlling the parameters λ and α . As a consequence, the variables x_i , y_i , and z_i will behave accordingly. However, regardless of the regimes, y_i is always positive and larger than zero for any positive non-null rates. Thus, this also implies that the mean-field equations do not predict a phase transition.

Action taken 2.2.— We changed the last paragraph of the Asymptotic analysis Section, better explaining our results. The paragraph reads as follows.

Note that, we can go from one regime to another by controlling the parameters λ and α . However, regardless of the regimes, y_i is always positive and larger than zero for any positive non-null rates in the three possible regimes, thus implying that the phase transition is not captured in the first-order approximation. Nonetheless, the transition was observed in our numerical experiments. Thus, we can conclude that the assumptions on the first-order mean-field approach are not enough to capture the essence of rumor models and correlations should be included.

COMMENT 2.3

- It would help to show explicitly that the same analysis applied to the SIRS model predicts the existence of a transition in that case.

Reply 2.3.— Thank you for your suggestion. We agree that it can be helpful.

Action taken 2.3.— We wrote an additional section in the Supplemental Material deriving the critical point for the SIRS model.

COMMENT 2.4

- Lambda_max is not defined in the main text (although it is defined in the Methods).

Isn't the largest eigenvalue of a k -regular graph equal to k ?

Reply 2.4.— Thank you for pointing that out. Also, the largest eigenvalue of a k -regular graph is approximately the average degree. In fact, this is true for homogeneous networks in general.

Action taken 2.4.— We defined the Λ_{\max} in the main text and mentioned that $\Lambda_{\max} \approx \langle k \rangle_k$ in homogeneous networks.

COMMENT 2.5

In addition, I have many suggestions for amendments to correct details in the manuscript that considerably reduce its readability.

- Starting from the abstract, the transition found in the MT model is denoted as second-order. A few lines before the mean-field approach used in previous papers is denoted as first-order. This is confusing, as the two "orders" are not related. I suggest to denote the transition as "continuous".

Reply 2.5.— Thank you for pointing that out. We agree that this terminology might be confusing.

Action taken 2.5.— We followed the reviewer's suggestion and changed every instance of the term "second-order phase transition" to "continuous phase transition".

COMMENT 2.6

- Brackets are used to denote averaging over realizations of the dynamical process and also averaging over the degree distribution. This is a bit confusing.

Reply 2.6.— Thanks for pointing that out. Despite being used in many network science papers, we do agree that it might generate some confusion.

Action taken 2.6.— We changed the notation of the expectation operator concerning the degree distribution from $\langle k \rangle$ to $\langle k \rangle_k$, avoiding confusion.

COMMENT 2.7

- The statement "causing impact on more than 50 years of investigation" is definitely excessive.

Reply 2.7.— Our rationale was that for many years it was thought that rumor models would not have phase transitions, which is the mean-field prediction. However, we agree that it might be excessive.

Action taken 2.7.— We removed that sentence in the abstract.

COMMENT 2.8

- Page 2, col. 2: weather → whether

Reply 2.8.— Thank you for pointing that out.

Action taken 2.8.— We corrected it.

COMMENT 2.9

- Page 3, beginning of col. 1: This limitation can be related to the non-exponential decay present in the sub-critical behavior"

“Non-exponential decay” suggests a temporal decay. If I understand correctly here a non-exponential decrease with lambda is intended. Is the non-exponential nature so important? Isn't the decrease (instead of the increase) the crucial point?

Reply 2.9.— Thank you for pointing that out. The important feature is the non-monotonic behavior of the lifespan. For instance, in disease spreading we would only have an increasing behavior as a function of λ , i.e., as we increase λ the lifespan increases, until we reach the critical point. However, here, in the Maki–Thompson model we have that the lifespan is non-monotonic where it is large for lower values of λ , then it decreases and increases again. This behavior is depicted in figures 3, 6, 7 and 8, for both the standard Maki–Thompson model (Fig. 3) and the modified model (figures 6, 7 and 8).

Action taken 2.9.— We changed the aforementioned sentence by

This limitation can be related to the non-monotonic behavior of the lifespan as a function of λ present in the sub-critical behavior.

COMMENT 2.10

- Page 4, col. 1: “To characterize the MT model, we can simulate the process beginning with many different initial conditions” What are the initial conditions used in Figs. 3, 6 and 8?

Reply 2.10.— Thank you for pointing that out. Here all the initial conditions have the same macro-state $\rho_Y = \frac{1}{N}$, i.e., it is always a single spreader. However, the initial spreader changes in the different runs.

Action taken 2.10.— We added the following sentence in the paragraph mentioned by the reviewer:

Note that the initial condition must correspond to a single spreader in an ignorant population to capture the transition. Thus, the initial spreader node changes in the different independent runs of our simulation.

COMMENT 2.11

- Fig. 3: Is the decay for small lambda close to $1/\lambda$ also in this case? Can you put a straight line with slope -1 in the figure?

Reply 2.11.— Yes, the decay in this case is also similar to $\tau \sim \lambda^{-1}$.

Action taken 2.11.— We included the curve $\tau \sim \lambda^{-1}$ in the lower λ regime.

COMMENT 2.12

- Fig. 4: The meaning of λ^ should be specified in the caption. What does $q(1)$ mean in the legend? Doesn't it mean that the value plotted is λ_{c} for which $q(1)=1$ in Eq. (4)? If this is the case, the indication “ $q(1)$ ” in the legend is misleading.*

Reply 2.12.— Thank you for pointing that out. The estimation λ^* was better described in the main text (please see also Comment 2.16). Regarding the estimation previously denoted by $q(1)$, the reviewer is correct, it is the value of λ in which $q(1) = 1$.

Action taken 2.12.— In the Fig. 4, we changed the legend explicitly mentioning the expression $q(1) = 1$. We improved the description of λ^* in the main text. Also, we included a brief description of λ^* and better explained the critical point calculated by $q(1) = 1$ in the caption of Fig. 4. The caption of Fig. 4 now reads

Comparison between analytical and Monte Carlo critical point estimations (λ_c). Results for random regular networks with $\langle k \rangle_k = 10$ and $\delta = 1$ and $N = 10^6$. The continuous line expresses the value of λ_c obtained as a solution of $q(1) = 1$, from Eq. (5). In contrast, the dashed line represents the naive approximation that accounts only for the probability that the next event is spreading or stifling. In the inset we present the comparison for the low α regime.

COMMENT 2.13

- Pag. 5, col. 1: where the lifespan increases as lambda increases → where the lifespan decreases as lambda increases

Reply 2.13.— Thank you for pointing that mistake.

Action taken 2.13.— We corrected the text as:

Moreover, we also observe an unexpected subcritical behavior, where the lifespan increases as λ decreases.

COMMENT 2.14

- Can you specify in detail the form of Eq. (4) for the random regular case ?

Reply 2.14.— Eq. (4) is a recurrent expression, and, despite our efforts, we were not able to find a closed expression or further simplify it. Not even after some structural assumptions.

COMMENT 2.15

- Fig. 5: what is the value of delta?

Reply 2.15.— Thank you for mentioning that. In all of our simulations for the $\delta > 0$ we used $\delta = 1$. Note that δ can be interpreted as the time scale of the process.

Action taken 2.15.— We added this information in the caption of Fig. 5 as

Critical point estimations of uncorrelated power-law networks. We plot λ_c as a function of N and for different values of γ and α , considering $\delta = 1$.

COMMENT 2.16

- Pag. 6, col. 1: how is λ^* derived? Does it come from Eq. (4) under some approximation?

Reply 2.16.— Thanks for the comment. We apologize for the confusion. This approximation is not related to Eq. (4). In fact, this approximation is based on the local behavior of a process starting from two connected spreaders. In this case, each spreader has $(\langle k \rangle - 1)$ ignorant neighbors to spread the rumor and a single neighbor to stop the process (the edge that connects the spreader and the stifier). So, the probability of spreading is $\frac{2\lambda(\langle k \rangle - 1)}{2\lambda(\langle k \rangle - 1) + 2\alpha}$, while the probability of stopping the rumor is $\frac{2\alpha}{2\lambda(\langle k \rangle - 1) + 2\alpha}$. Thus, these two probabilities will be the same at $\lambda^* = \frac{\alpha}{(\langle k \rangle - 1)}$. Finally, from this approximation one would expect that, if $\lambda > \lambda^*$ the process can spread to a fraction of the population, while if $\lambda < \lambda^*$ it should be constrained to a finite fraction of nodes.

Action taken 2.16.— We added the following explanation below its definition:

Moreover, λ^* is the rate at which spreading and stifling have the same probability. So, from this approximation, one would expect that, if $\lambda > \lambda^*$, the process can spread to a fraction of the population, while if $\lambda < \lambda^*$ it should be constrained to a finite fraction of nodes.

COMMENT 2.17

- Fig. 6: How has the orange line been derived? Fitting to what? The number of digits of the exponent is excessive.

Reply 2.17.— The orange curve was obtained as a fitting from the lifespan obtained using Monte Carlo simulations, the gray curve in Fig. 6. The main purposes of showing this curve are to show that our approximations are reasonable and also to guide the eye.

Action taken 2.17.— We changed the Fig. 6 and its caption, reporting only two decimals of precision. Now the last sentence of the caption of Fig. 6 reads as

The blue curve (dot dashed line) follows $\tau_f \sim \lambda^{-1}$ and the orange curve (dashed line) follows $\tau_f \sim \lambda^{-0.88}$, obtained from a fitting of the lifespan obtained using Monte Carlo simulations (the gray curve).

COMMENT 2.18

- Eq. (6): $k \rightarrow < k >$

Reply 2.18.— Thank you for pointing that out.

Action taken 2.18.— We corrected it.

COMMENT 2.19

- Pag. 7, col. 1: "the time to reach the absorbing state, T_{abs} . This quantity was called tau in Eq. (2).

Reply 2.19.— Thank you for pointing that out.

Action taken 2.19.— We emphasize that τ obtained by Monte Carlo simulations should converge to the theoretical value T_{abs} . We added the following explanation.

The subcritical regime can be characterized in terms of the time to reach the absorbing state, T_{abs} . We remark that τ in Eq. (3) should converge to T_{abs} .

COMMENT 2.20

- Pag. 7: $A +$ is missing in Eq. (9).

Reply 2.20.— Thank you for pointing that out.

Action taken 2.20.— We corrected it.

COMMENT 2.21

- Pag. 10, col. 1, after Eq. (12): ρ_Y is the quasi-stationary distribution $\rightarrow \rho_Y$ is the quasi-stationary density.

Reply 2.21.— Thank you for pointing that out.

Action taken 2.21.— We replaced the term “quasi-stationary distribution” to “the quasi-stationary density”.

COMMENT 2.22

- *Supplementary Information is not mentioned in the manuscript.*

Reply 2.22.— Thank you for pointing that out.

Action taken 2.22.— We included three references to the Supplemental Material. First, in Section II.C.1, page 6, first column, we added one reference to the Supplemental Material that reads as

In Fig. 4 we compare the solution of $q(1)$ with Monte Carlo critical point estimations for random regular networks with $\langle k \rangle = 10$ for both $\delta = 0$ and $\delta = 1$ (see Section IV A for the simulation details and the Supplemental Material for the individual susceptibility curves and critical point estimations).

Second, at the end of the same section we also mentioned the results in the Supplemental Material as

These results are summarized in Fig. 5 (please see the Supplemental Material for the individual susceptibility curves and critical point estimations).

And third, at the end of the Results section, we included:

A similar asymptotic analysis was made for the SIRS model in the Supplemental Material, for the sake of comparison.